# A Novel ICESat-2 Signal Photon Extraction Method Based on Convolutional Neural Network

Wenjun Qin [1], Yan Song [1,*], Yarong Zou [2,3], Haitian Zhu [2,3] and Haiyan Guan [4]

1   School of Geography and Information Engineering, China University of Geosciences, Wuhan 430074, China; qinwenjun@cug.edu.cn
2   Key Laboratory of Space Ocean Remote Sensing and Application, Ministry of Natural Resources, Beijing 100081, China; zyr@mail.nsoas.org.cn (Y.Z.); zht@mail.nsoas.org.cn (H.Z.)
3   National Satellite Ocean Application Service, Beijing 100081, China
4   School of Remote Sensing & Geomatics Engineering, Nanjing University of Information Science and Technology, Nanjing 210044, China; guanhy.nj@nuist.edu.cn
*   Correspondence: songyan@cug.edu.cn

**Abstract:** When it comes to the application of the photon data gathered by the Ice, Cloud, and Land Elevation Satellite-2 (ICESat-2), accurately removing noise is crucial. In particular, conventional denoising algorithms based on local density are susceptible to missing some signal photons when there is uneven signal density distribution, as well as being susceptible to misclassifying noise photons near the signal photons; the application of deep learning remains untapped in this domain as well. To solve these problems, a method for extracting signal photons based on a GoogLeNet model fused with a Convolutional Block Attention Module (CBAM) is proposed. The network model can make good use of the distribution information of each photon's neighborhood, and simultaneously extract signal photons with different photon densities to avoid misclassification of noise photons. The CBAM enhances the network to focus more on learning the crucial features and improves its discriminative ability. In the experiments, simulation photon data in different signal-to-noise ratios (SNR) levels are utilized to demonstrate the superiority and accuracy of the proposed method. The results from signal extraction using the proposed method in four experimental areas outperform the conventional methods, with overall accuracy exceeding 98%. In the real validation experiments, reference data from four experimental areas are collected, and the elevation of signal photons extracted by the proposed method is proven to be consistent with the reference elevation, with $R^2$ exceeding 0.87. Both simulation and real validation experiments demonstrate that the proposed method is effective and accurate for extracting signal photons.

**Keywords:** ICESat-2; signal photon extraction; photon data transformation; GoogLeNet; CBAM




## 1. Introduction

As a new-generation surface exploration satellite developed by NASA, ICESat-2 carries an Advanced Topographic Laser Altimeter System (ATLAS), which enables it to calculate the distance between the satellite and the Earth surface by measuring the propagation time of the laser pulse, deriving the change in land topography and providing high-precision surface elevation information [1]. Carrying the ATLAS at an orbital altitude of 500 km, ICESat-2 operates at 532 nm and emits three pairs of beams with a spacing of 3.3 km and a distance of 90 m between the two pairs of beams. Each pair of beams consists of a strong beam and a weak beam, with the intensity of the strong beam being four times greater than that of the weak beam. For intensive sampling and the efficient capture of elevation changes, each beam is designed with a diameter of 17 m and a sampling interval of 0.7 m along the track. Due to its exceptional ability to provide highly precise surface elevation data, the ICESat-2 photon data has been utilized in a diverse range of current research studies. In terms of shallow sea bathymetry inversion, photon data can offer numerous high-precision

and extensive coverage bathymetric control points [2–7]. Additionally, in forested areas, photon data are also a reliable source for canopy height information over large regions, which is conducive to various studies such as surface biomass estimation [8–12]. In urban areas, photon data can also be utilized to provide information about the height of urban buildings, which assists in the detection of changes in urban areas [13,14]. In polar and alpine regions, photon data can also be applied to detect the elevation changes of alpine and polar glaciers [15–19].

The ATLAS system receives not only its own impulse signals but also noise caused by atmospheric scattering, solar radiation, and instrument artifacts. The initial collected photon data exhibits a significant amount of random noise that is distributed around the signal photons. Only after removing these random noises to get the clear signal photons can the data be utilized for various applications. As for the ICESat-2 photon data, the photon density within the neighborhood of a signal photon is significantly higher than that of a noise photon. Therefore, some conventional algorithms have been developed for noise photon removal, most of which are derived from the threshold segmentation algorithms based on local density or local distance. Among them, the most used one is Density-Based Spatial Clustering of Applications with Noise (DBSCAN) [20]. In order to include more signal photons in the search neighborhood, the circular neighborhood is commonly replaced by an elliptical neighborhood when the DBSCAN algorithm is applied [21–26]. Chen et al. [21] proposed an adaptive variational ellipsoidal filtering bathymetry method which processes histogram statistics in the depth direction to separate surface photons from bottom ones, and adaptive variable ellipsoid filters allow the method to extract more signal photons at deeper regions. Leng et al. [22] adopted a kernel density statistical approach to separate land and water, with the elliptical neighborhood flexibly rotated to accommodate more signal photons. This method can somewhat increase the local density gap between signal and noise photons. Nan et al. [23] proposed a method that combines local denoising with global denoising to effectively eliminate outliers, which improves the denoising accuracy by means of secondary denoising. Yang et al. [24] proposed backward elliptic distance (BED) to solve the signal photons fluctuate problem by rotating the elliptic domain, thus minimizing the points in the neighborhood. This method can obtain good results in mountainous areas. Zhang et al. [25] proposed an algorithm which uses a genetic algorithm to find the optimal denoising thresholds. As opposed to using empirical thresholds, this method allows for adaptive selection of thresholds based on the real situation. Zhu et al. [26] applied the Ordering Points to Identify the Clustering Structure (OPTICS) algorithm, which is similar to the DBSCAN algorithm, during processing procedure. But, compared with the DBSCAN algorithm, OPTICS is less sensitive to data parameters.

Despite this, all the conventional methods above, which are based on local density or local distance, only utilize a feature where the overall local density of signal photons is much denser than that of the noise photons. The position and shape features of the photons in the neighborhood of the signal and noise photons are not fully exploited. Meanwhile, there are also signal photons with different local densities in the same data. Such a data local point density histogram does not conform to the typical bimodal distribution, but there may be multi-peak distribution. Therefore, it is difficult to separate all the signal and noise well using the threshold segmentation methods.

The ICESat-2 photon data are morphologically similar to a 3D point cloud in that the data patterns are all scattered. There is some similarity between the two in terms of data analysis. Nowadays, deep learning methods have been widely used for 3D point cloud denoising. These existing methods can be divided into two categories, one being the network structure based on the PointNet framework [27–30]. This type of PointNet framework takes the 3D coordinates of the point cloud as the input to obtain the feature vector through a multilayer perceptron network, feature transformation network, and maximum pooling layer. Then, the classifier is used to perform classification based on these features. The other category converts the 3D point cloud data into voxel data or projects them onto a 2D depth map. Then, the Convolutional Neural Network (CNN) model is

used for feature learning and classification [31–34]. Theoretically, both types of network structures can make full use of the position, shape, geometry, and other features of the 3D point cloud data. However, the amount of geometric feature information of ICESat-2 photons is significantly reduced compared with the 3D point cloud data. Consequently, direct application of CNNs designed for 3D point clouds to the ICESat-2 photons is challenging to achieve accurate results. Therefore, deep learning methods have not yet been widely applied to ICESat-2 photons.

To solve these problems, a novel CNN-based ICESat-2 signal photon extraction method is proposed in this article. Firstly, it transforms each photon into a 2D image with a specific aspect ratio to preserve feature information in its neighborhood. Subsequently, a GoogLeNet fused with a CBAM is trained from the photon image data, which specializes in capturing the detailed original feature information of the photon image data across different scales and achieves a more comprehensive feature representation. As a result, not only local densities features, but also position and shape features of the signal and noise photons are well learned, which enables a constant increase in the feature gap between signal photons and noise photons. In addition, the CBAM [35] enhances the network's ability to focus more on learning the crucial features and improves its discriminative ability. Ultimately, the proposed method can overcome the obstacles encountered when extracting signal photons using the conventional approach, which can be proven in both simulation and real reference experiments.

This paper consists of six sections. Section 1 primarily introduces basic information about ICESat-2 photon data and the limitations of conventional methods. Section 2 describes the basic information of the study areas and data. Section 3 specifically comprehensively describes the method proposed in this paper, which is mainly divided into two parts: photon data transformation and CNN network modeling. Section 4 presents the experimental results comparing the proposed method with the conventional methods to demonstrate the superiority of the proposed method. Section 5 conducts a comparative analysis between classical CNN models and those incorporating an attention mechanism. Finally, Section 6 concludes by summarizing the proposed method presented in this paper.

## 2. Data

### 2.1. Experimental Area and Data

Figure 1 presents an overview of the experimental area for this paper. Four distinct sets of ICESat-2/ATL03 data are utilized in this study, which are shown in Table 1. Experimental Areas A and B are both located in the Cibola National Forest. Specifically, experimental area A is located in the central Sedgwick Mountains of New Mexico, while experimental area B is located in the northern San Mateo Mountains of New Mexico. Covered by forests, the density of ground photons in such areas are much denser than that of vegetation canopy photons. Experimental area C is located in close proximity to the Ramrod Island Reef in the Florida Keys. By utilizing water surface and bottom signal photons, topographical information of the shallow water bottom can be obtained. However, the signal photon density at the water surface is much denser than at the water bottom due to the gradual laser energy propagation inside the water. The conventional methods often fail to fully extract the signal photons from the water bottom, resulting in a lack of topographical information [24–26]. Experimental area D is located near Sanostee, within the state of New Mexico. The land surface in this area is essentially devoid of feature cover and the signal photon composition is unitary in the data from this area, which are primarily utilized to test the denoising ability of different methods on simple data.

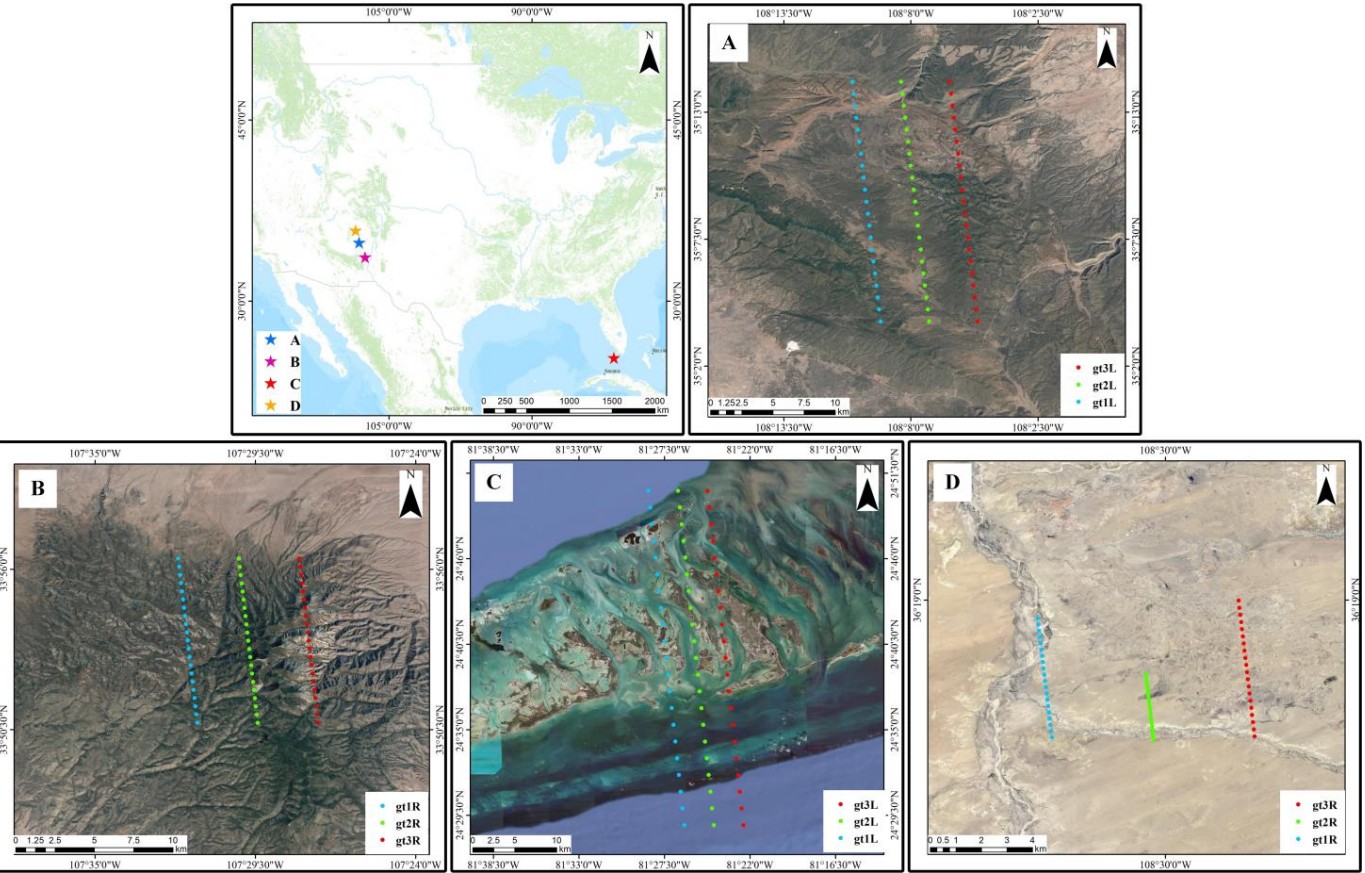

**Figure 1.** Overview of the experimental areas A, B, C and D.

**Table 1.** Information on ICESat-2/ATL03 data used.

| Experimental Area | ICESat-2/ATL03 | Data Acquisition Time | Track Used |
|:---:|:---:|:---:|:---:|
| A | ATL03_20200914131032_12400802_005_01.h5 | 14 September 2020 14:13:10 | gt1L/gt2L/gt3L |
| B | ATL03_20191114034331_07370502_005_01.h5 | 14 November 2019 3:43:31 | gt1R/gt2R/gt3R |
| C | ATL03_20190807063856_06140401_005_01.h5 | 7 August 2019 6:28:56 | gt1L/gt2L/gt3L |
| D | ATL03_20221111233317_07981702_006_01.h5 | 11 November 2022 23:33:17 | gt1R/gt2R/gt3R |

### 2.2. Real Validation Data

In order to validate the correctness of the signal photons obtained by the proposed method, a Digital Terrain Model (DTM) raster, Digital Elevation Model (DEM) raster, and bathymetric raster are used to validate. The DTM raster is obtained from the Goddard's Li-DAR, Hyperspectral and Thermal Imager (G-LiHT) dataset with a spatial resolution of 1 m (https://glihtdata.gsfc.nasa.gov/, accessed on 7 December 2023). The G-LiHT dataset [36] was collected by the NASA science team using the G-LiHT system. G-LiHT is a portable airborne imaging system containing LIDAR, hyperspectral, and thermal imager sensors that simultaneously map the composition, structure, and function of terrestrial ecosystems. The bathymetric raster is obtained from the 2018–2019 NOAA NGS Topobathy Lidar DEM Hurricane Irma project [37] which are collected by three Riegl systems and then used to create raster data with a spatial resolution of 1 m (https://coast.noaa.gov/htdata/raster2 /elevation/NGS_FL_Topobathy_PostIrma_MiamiToMarquesas_2019_9060/, accessed on 7 December 2023). The DEM raster is obtained from The Terra Advanced Spaceborne

Thermal Emission and Reflection Radiometer (ASTER) Global Digital Elevation Model (GDEM) Version 3 [38]. The geographic coverage of the ASTER GDEM extends from 83° north to 83° south. Studies to validate and characterize the ASTER GDEM confirm that accuracies for this global product are 20 m at a 95% confidence for vertical data and 30 m at a 95% confidence for horizontal data.

### 2.3. Trianing Dataset

Since there is no universally applicable training dataset for ICESat-2 signal photon data, in order to train the model and test the performance of the proposed method, we designed simulation data for the training dataset.

The data construction process is as follows: first, for a given original dataset, we manually extract photons with high confidence as signal photons using validation data such as DEM as a reference. These photons are used as signals in the training dataset. Subsequently, in order to simulate the real situation where the noise levels in different photon data are different, we introduced Gaussian white noise with different signal-to-noise ratios (SNR) into the above extracted signal photons. In practice, Gaussian white noise with SNRs of 60, 70, 80, and 90 (dB) are added to the signal photons as noise in the training dataset, respectively. Eventually, four simulation data are generated from one original data, which are used to simulate the original photon data with different noise levels in the real situation and at the same time. With the powerful learning and generalization ability of the neural network, the network model can discriminate the abilities of different original data noise in the real situation. All simulation data are divided into two parts: a training dataset and a validation dataset. The training dataset is used to train the proposed model. The validation dataset is used to compare the denoising performance of different methods. Finally, real data validation experiments are conducted and the results further illustrate the correctness of the proposed method in this paper.

## 3. Methods

As illustrated in Figure 2, the proposed method consists of four main steps. Firstly, ICESat-2/ATL03 datasets are collected in various experimental areas, and the information in the along-track direction, including elevation and incident angle, are extracted to generate the photon distribution map. Secondly, transformed photon images are derived through a three-step process involving image size determination, feature information extraction, and feature information combination. Subsequently, each photon and its neighborhood photons are transformed into a photon image. Thirdly, during the CNN network training process, the CNN network model used in the article is the GoogLeNet model with CBAM module. Training samples of the transformed photon images are utilized to train the model. Finally, the results are validated by both simulation and real reference experiments. Simulation validation experiments utilize multiple datasets of different SNR levels, which are then denoised by the proposed method to demonstrate its effectiveness. In the real validation experiments, the extracted information from denoised signal photons is compared with the reference data to further demonstrate the effectiveness of the proposed method.

### 3.1. Photon Data Transformation

Due to the inherent properties of ICESat-2 photon data, only two types of information can be derived from the photon data, namely the along-track distance (X coordinates) and elevation values (Z coordinates). The conventional threshold segmentation method works well only if the neighborhood range and direction fit the distribution of signal photons and the local density of signal photons is highly concentrated. However, the distribution of real photon data is often more complicated. When the neighborhood is inconsistent with the trend of signal photons, such photons are often misclassified as noise. Meanwhile, if there are two groups of signal photons but with different degrees of clustering, signal photons with relatively low clustering will be misclassified as noise photons. Besides, the noise

photons near the signal photons will be misclassified as signal photons because of the large number of signal photons in its neighborhood.

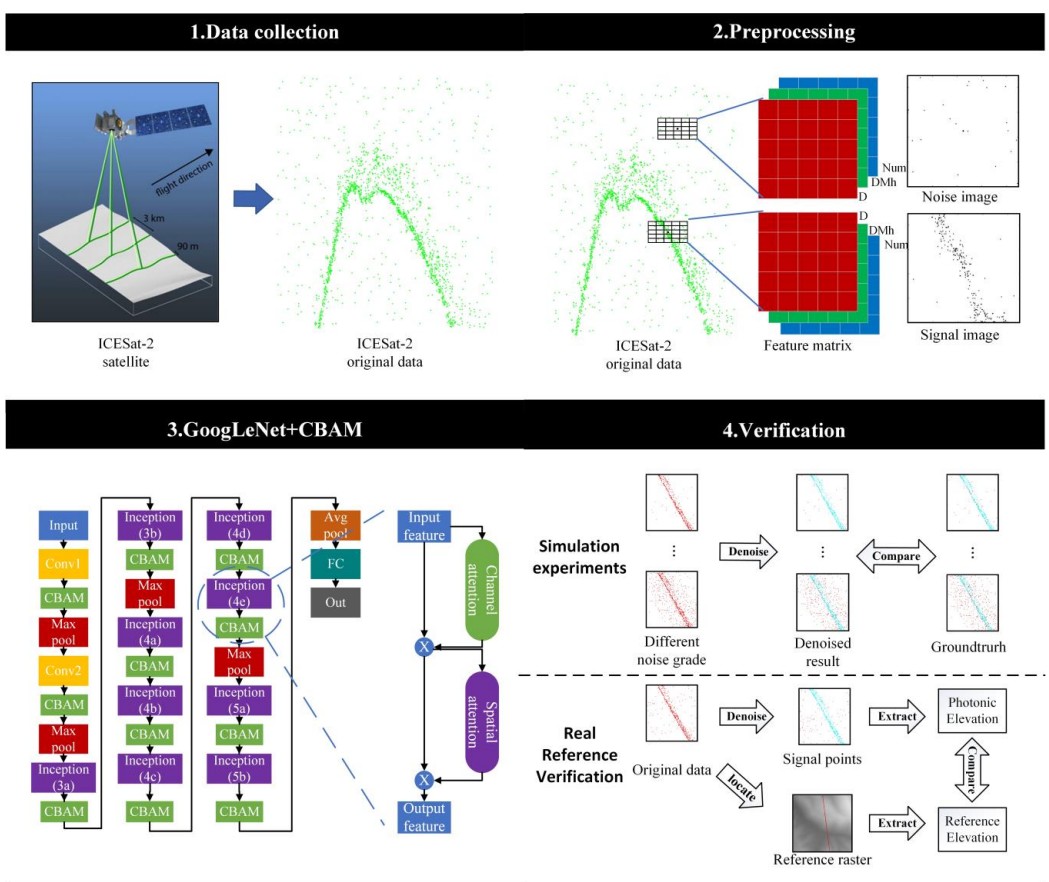

**Figure 2.** Method flow chart.

To solve these problems, the initial photon and its adjacent photons are transformed into a 2D image with dimensions determined by the distribution range of the signal photons' along-track distance and limited fluctuation elevation. The overall process is shown in Figure 3. Typically, the distance along the track direction is much greater than the elevation direction, resulting in a transformation range that is a rectangular domain rather than a normal square domain. It allows for good description of the location and shape distribution of photons within each photon neighborhood in the along-track distance direction, as well as good separation of signal and noise photons in the elevation direction, ultimately making the image samples more distinguishable from each other.

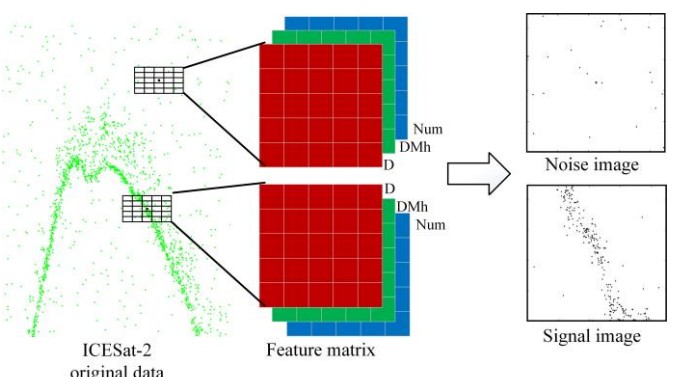

**Figure 3.** Schematic diagram of the photon transformation process.

The feature information of all photons in the neighborhood is transformed into a feature matrix, wherein three types of features are calculated: the number of photons (*Num*), the mean deviation of photon elevations from that of the center photon (*DMh*), and the distance between each photon to the center photon (*D*).

$$\begin{cases} Num = Count(p_1, p_2, \ldots, p_i, \ldots, p_k) \\ DMh = \frac{\sum_{i=1}^{k} H_{p_i}}{k} - H_{p_0} \\ D = \sqrt{\left(L - L_{p_0}\right)^2 + \left(H - H_{p_0}\right)^2} \end{cases} \tag{1}$$

where $p_i$ is the *i*th photon in one grid of the 2D image; Count is the number statistics functions; $H_{p_i}$ is the elevation of each photon; $H_{p_0}$ is the elevation of the photon at the center of the grid $p_0$; $L$ and $H$ are the along-track distance and the elevation of the center of each cell; and $L_{p_0}$ and $H_{p_0}$ are the along-track distance and elevation of the grid center photon $p_0$. These three feature matrices are first normalized separately and they are combined to construct a three-band image, then the transformation is completed.

### 3.2. CNN Model

The 2D transformed photon image can be effectively utilized to train the CNN. In the article, GoogLeNet [39] serves as the backbone network for deep learning. Previous studies of network structures aimed to improve the training results by increasing the depth (layers) of the network, while excessive layers may lead to negative effects, such as overfitting, gradient disappearance, gradient explosion, etc. [39].

Therefore, the Inception module is proposed to enhance the training results by optimizing computational resource utilization and extracting more features with the same amount of computational workload. By incorporating convolutional kernels of varying sizes within a single layer, the Inception module expands the "width" of the network model and improves performance in capturing detailed original feature information across different scales, ultimately achieving more comprehensive feature representation. In the signal photon images, there are variations in shape and position features among different Earth surface scenes. The Inception module is capable of extracting the distinct features of the signal photon images, allowing the network to discriminate them from the noise photon images. The schematic of the Inception module is depicted in Figure 4:

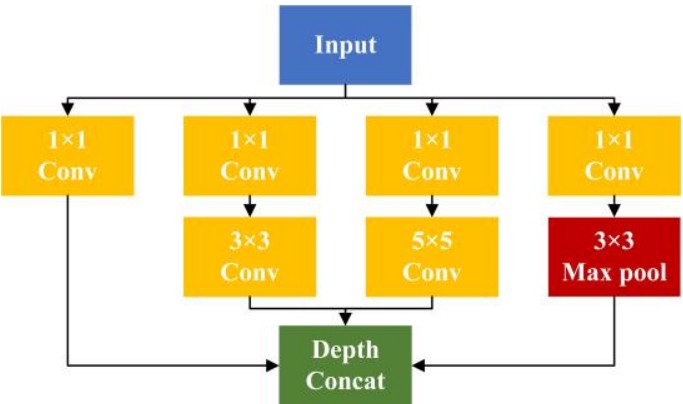

**Figure 4.** Schematic diagram of the Inception module [39].

To enhance the ability of GoogLeNet to learn more effective features from the training sample images, the CBAM is further introduced into the network backbone. The CBAM, consisting of the Channel Attention Module (CAM) and the Spatial Attention Module (SAM), focuses on "attention" to improve the performance of the network by allowing the network to learn which information to emphasize or suppress, thereby facilitating effective information flow [35]. Figures 5 and 6 show the specific structures of CAM and

SAM, respectively. Specifically, the CAM identifies meaningful channels in input features, while SAM determines relevant features at spatial positions. The initial input features are sequentially passed through the CAM and SAM modules, during which the output features need to be multiplied pixel by pixel with the input features, resulting in the final CBAM-weighted features.

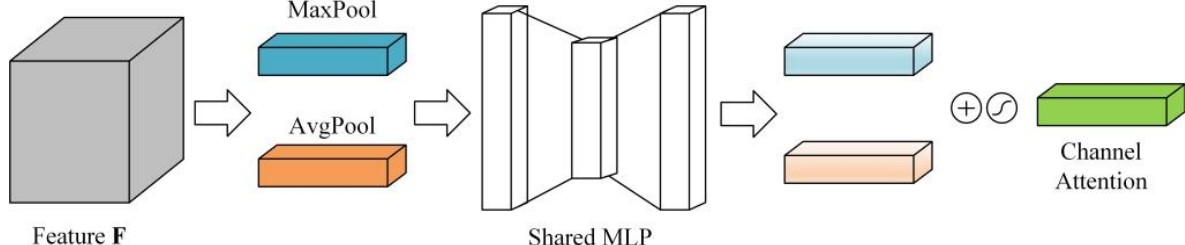

**Figure 5.** Schematic diagram of the CAM module [35].

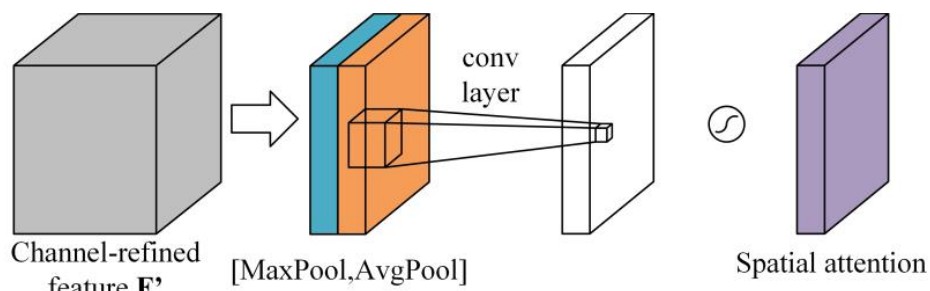

**Figure 6.** Schematic diagram of the SAM module [35].

In this study, as shown in the third part of Figure 2, the primary network structure of GoogLeNet is applied to construct the network, with the incorporation of the CBAM module. The first and second parts of the network consist of both convolutional (Conv1 and Conv2 in Figure 2) and max-pooling layers (Max pool in Figure 2), with one and two convolutional layers, respectively. The CBAM modules are introduced after these convolutional layers. The third, fourth, and fifth parts of the network are predominantly composed of Inception modules, with the numbers 3, 4, and 5 corresponding to them. Each Inception module is succeeded by a CBAM and a max-pooling layer. As the output layer, the sixth part consists of an average-pooling layer (Avg Pool in Figure 2) and a fully connected layer (FC in Figure 2).

*3.3. Validation and Evaluation*

The validation process in this paper is divided into two main parts: simulation experiment validation and real reference validation. For the simulation experiment validation, the denoised results of the simulation data are evaluated by four metrics: Precision, Recall, Overall Precision (OA), and Kappa coefficient. For the real reference validation, four sets of ICESat-2 photon data are collected for real validation. The denoised results are compared with real measurements to validate the effectiveness of the proposed method. For the real data validation, RMSE and $R^2$ are utilized and the mean absolute error (MAE) and mean relative error (MRE) are also calculated.

**4. Experimental Process and Results**

*4.1. Experimental Process*

The whole experimental process is mainly divided into two parts. The first part is the simulated data construction and photon data transformation. The simulated data of these four experimental areas are constructed using the method in Section 3.3. Twelve tracks of data from the four experimental areas incorporate noise of four different SNR

levels, and forty-eight pieces of simulated data are finally obtained. Then, all the photons in these simulated data are converted to photon images using the photon data transformation method in Section 3.1. The second part is the training and prediction of the network model. Since the four experimental areas in this paper are categorized into three main feature types, forest (A and B), shallow sea (C), and bare soil (D), the three network models will be trained separately according to the feature types in the actual experimental process. Then, thirty-two of the forty-eight simulated data obtained above are used as training and validation samples for training the network model in Section 3.2. In total, there are 182,176 sample images in experimental area A and experimental area B, 278,373 photon images in experimental area C, and 40,512 photon images in experimental area D. The ratio of the training set to the validation set is 8:2. The number of positive samples and negative samples is the same. Table 2 demonstrates the hyperparameter settings in network training. Network models are trained with the Adam optimizer using the CrossEntropyLoss loss function. Figures 7 and 8 show the change curves of the accuracy value and the loss value of the first two network models, respectively. Finally, the three trained models are used for denoising the remaining sixteen simulated data to get the results, which will be compared with the results of other denoising methods. There are three conventional methods that will be used to compare with the proposed method in the simulation and real validation experiments: the DBSCAN [20], OPTICS [26], and BED [24]. The first two trained network models are also used to denoise the raw data of three experimental areas. After obtaining the signal photons, the elevation information of the signal photons is calculated and compared with the real validation data.

**Table 2.** Information about the training hyperparameters.

| Training Hyperparameter | Setting |
| --- | --- |
| Initial learning rate | 0.001 |
| Learning rate change modality | ExponentialLR |
| Learning rate change rate | 0.98 |
| Batch size | 2 |
| Optimizer | adam |
| Training epoch | 20 |
| Training to validation | 8:2 |

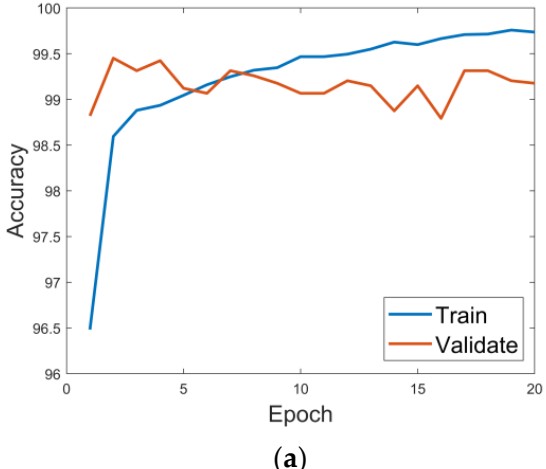

(**a**)

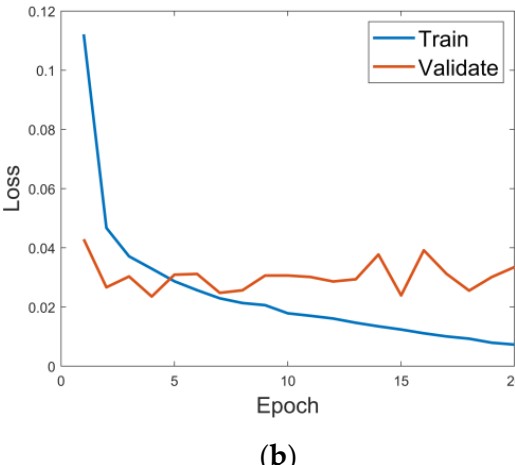

(**b**)

**Figure 7.** The training process of network model in experimental areas A and B. (**a**) Change curve of accuracy; (**b**) change curve of loss.

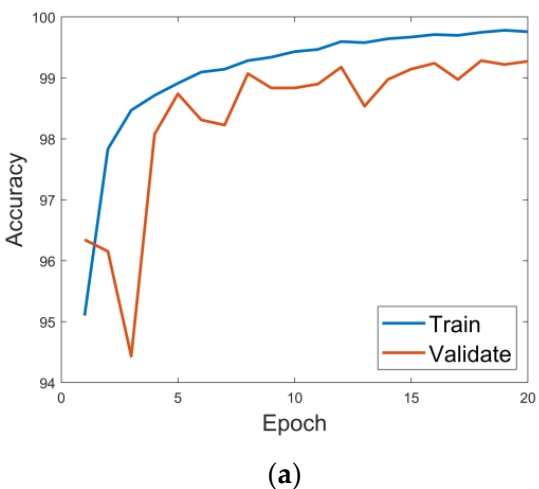

(a)

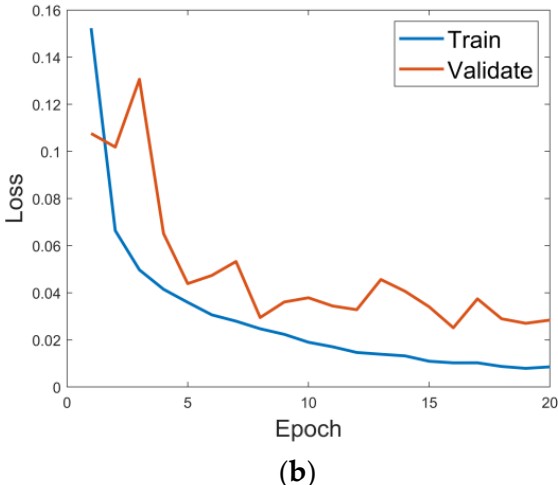

(b)

**Figure 8.** The training process of the network model in experimental area C. (**a**) Change curve of accuracy; (**b**) change curve of loss.

### 4.2. Typical Photon Classification Results

Figure 9 presents the signal photon extraction results of the conventional method and proposed method. For conventional methods, there are two types of photons that cannot be classified well. The first type pertains to the omission of signal photons with low local density. In Figure 9, the center photon in window A and window B are both signal photons, and the center photon in window D is a common noise photon. Notably, the quantity of photons within window A significantly exceeds that in windows B. So, the threshold is typically biased for the conventional algorithm, which makes it easy to misclassify the signal center photon in window B as noise photon when using conventional methods. However, the distribution characteristics of neighboring photons differ between the center signal photons in window A and window B and the noise photon in window D. The noise photons in window D are randomly distributed, while the signal photons in windows A and B are distributed along the water surface or underwater terrain. They are significantly different. Additionally, as presented in Figure 10, the images transformed using the proposed method also highlight these differences. Therefore, the signal photon in window B with a lower local density can be readily discriminated from the noise photon in window D. The proposed method can get better results in this situation.

Meanwhile, the second type is about the noise photons close to the signal photons that are misclassified. In Figure 9, the noise photon in window C is in close proximity to the signal photons. For such a noise photon, there are lots of signal photons in its neighborhood. The difference between the number of photons surrounding the noise photon in window C and that surrounding the signal photon in window A is insignificant, making it difficult to distinguish between them using conventional methods. But, the transformed photon image contains nearby signal photons along with the noise photon. As presented in Figure 10, there is a significant difference in the position of their corresponding signal photons between the noise photon in window C and signal photon in window A. Taking advantage of this, the noise photon in window C would not be misclassified in the proposed method.

### 4.3. Simulation Experimental Results

Experimental area A is covered by forests. Since the ATLAS system digitally records all the laser echo signals from the canopy to the ground, the shape of the photon data distribution in the forested area is largely determined by the vertical structure of the tree canopy [40]. A coniferous forest with low density dominates this area, and Figure 11 illustrates the overall distribution of photon data in this area. Different photon densities make it challenging to differentiate between ground signal photons and canopy signal

photons. As depicted in Figure 12, while all three conventional methods are capable of extracting ground signal photons with higher local density (in the blue box), they tend to miss some canopy signal photons (in the green box), resulting in fair accuracy but low recall. The proposed method takes into account the shape and position properties of signal photons in the neighborhood of different local densities, resulting in more accurate extraction results for both ground and canopy signal photons. Meanwhile, Table 3 illustrates the denoising results of different denoising methods for photon data in different SNR levels. The conventional methods can achieve high overall accuracy only under photon data in low SNR levels. Under the photon data in higher SNR levels, the local density difference between the signal photons and the noise photons becomes smaller, and thus the denoising effect of the traditional methods becomes worse. In contrast, the proposed method obtains high denoising results under photon data on all SNR levels, which proves that the proposed method is more adaptable. Figure 13 demonstrates the curves of the results of the four methods for the four validation metrics under different SNRs in experimental area A. Compared to the conventional methods, the denoising results obtained by the proposed method (red line) are not affected by the different noise level, and the accuracy of the results are all better.

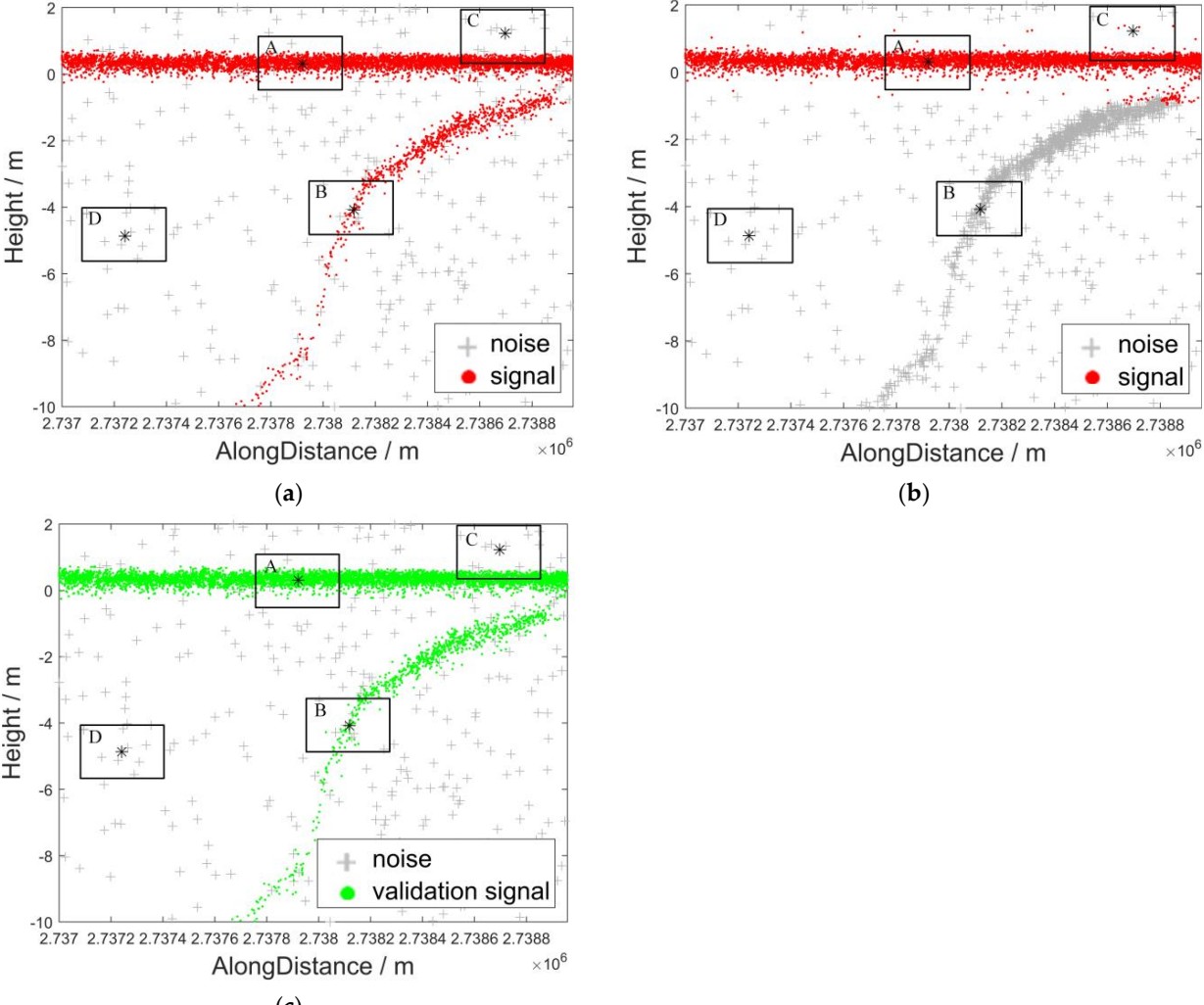

**Figure 9.** Typical photon neighborhood of photon A, B, C and D and comparison of denoised results. (**a**) The denoised result of the proposed method (SNR = 80 dB); (**b**) the denoised result of DBSCAN (SNR = 80 dB); (**c**) validation.

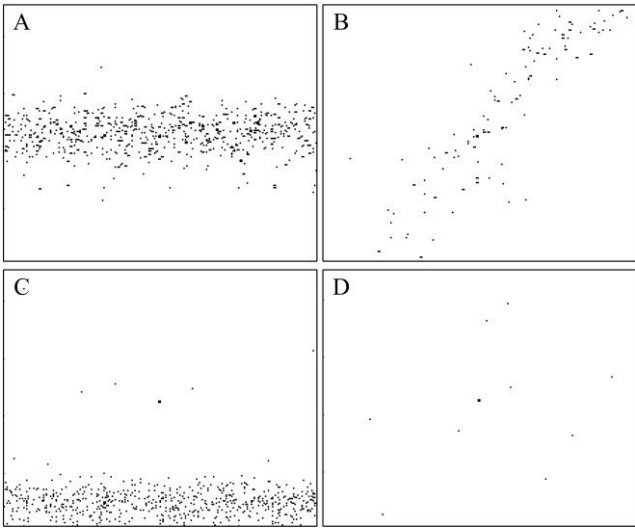

**Figure 10.** Photon images of typical photons A, B, C, and D in Figure 9.

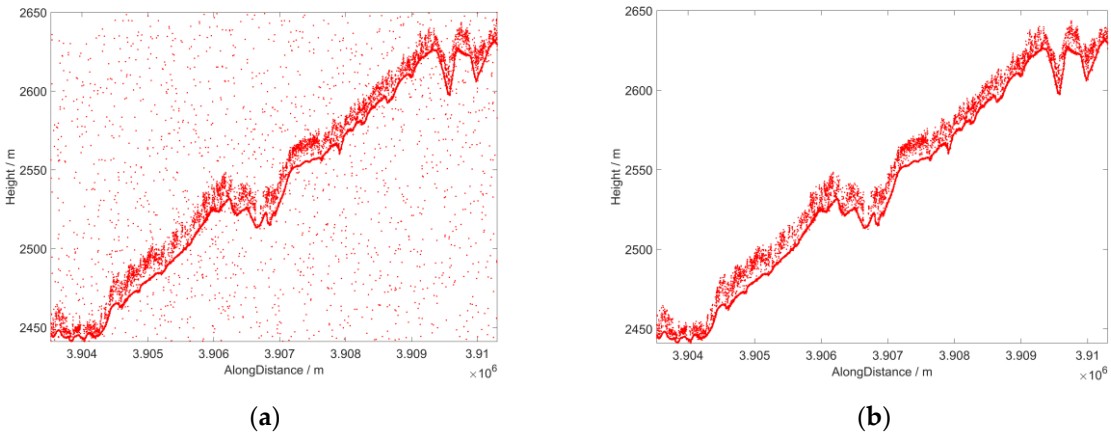

(**a**)                         (**b**)

**Figure 11.** Overall distribution of photon data in experimental area A. (**a**) Original photon data; (**b**) signal photon data.

**Table 3.** Results of the gt2L track in experimental area A.

|                      | SNR (dB) | Precision | Recall  | OA      | Kappa   |
|----------------------|----------|-----------|---------|---------|---------|
| The proposed method  | 60       | 99.97%    | 98.96%  | 99.47%  | 98.93%  |
|                      | 70       | 99.94%    | 98.94%  | 99.44%  | 98.88%  |
|                      | 80       | 99.82%    | 98.89%  | 99.35%  | 98.71%  |
|                      | 90       | 99.50%    | 98.72%  | 99.11%  | 98.22%  |
| DBSCAN               | 60       | 99.69%    | 91.59%  | 95.65%  | 91.30%  |
|                      | 70       | 98.95%    | 93.14%  | 96.08%  | 92.15%  |
|                      | 80       | 95.72%    | 94.15%  | 94.97%  | 89.94%  |
|                      | 90       | 82.82%    | 97.62%  | 88.68%  | 77.37%  |
| OPTICS               | 60       | 99.84%    | 50.54%  | 75.23%  | 50.46%  |
|                      | 70       | 98.95%    | 72.13%  | 85.68%  | 71.36%  |
|                      | 80       | 98.47%    | 72.81%  | 85.84%  | 71.68%  |
|                      | 90       | 93.44%    | 75.33%  | 85.02%  | 70.04%  |
| BED                  | 60       | 99.63%    | 82.85%  | 91.27%  | 82.54%  |
|                      | 70       | 98.62%    | 83.35%  | 91.09%  | 82.19%  |
|                      | 80       | 95.02%    | 84.58%  | 90.07%  | 80.14%  |
|                      | 90       | 80.85%    | 88.14%  | 83.63%  | 67.27%  |

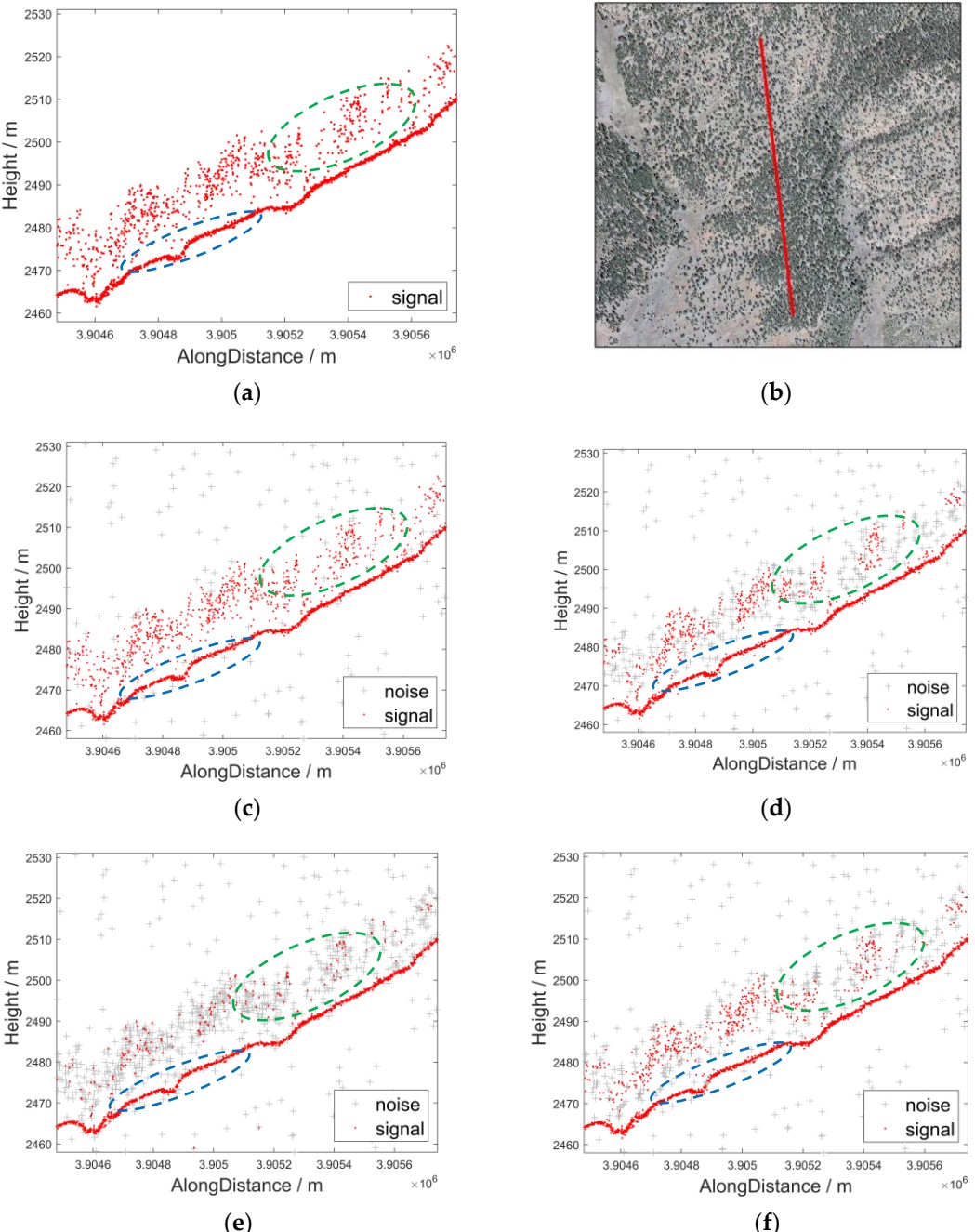

**Figure 12.** Comparison of the details of the gt2L track results in the experimental area A (SNR = 70 dB). (**a**) validation; (**b**) optical remote sensing image; (**c**) the proposed method; (**d**) DBSCAN; (**e**) OPTICS; (**f**) BED.

In experimental area B, there is a higher density in the forest area compared with experimental area A. The overall distribution of the photon data in this area is shown in Figure 14 and the noise density is not very high. But, the signal photon density is not the same at different locations (steep slopes compared with gentle areas). It is challenging to extract both ground signal photons and canopy signal photons well. As represented in Figure 15, the DBSCAN algorithm performs best in extracting most of the signal photons among the conventional methods. However, there are still some omissions existing in the canopy area (in the green box) and on the ground (in the blue box). As for the other conventional methods, the omissions become greater. The forested terrain and canopy cannot be well extracted simultaneously. Meanwhile, the proposed method outperforms

others. Both ground and canopy signal photons are extracted completely, leading to the highest denoising accuracy. And in Table 4, it can be clearly observed that the denoising accuracy of traditional methods decreases significantly with increasing SNR levels, which does not occur with the proposed method in this paper. Figure 16 demonstrates the curves of the results of the four methods for all validation metrics under different SNRs in experimental area B. Compared to the conventional methods, the denoising results obtained by the proposed method (red line) are not affected by the different noise level, and the accuracy of the results are all better.

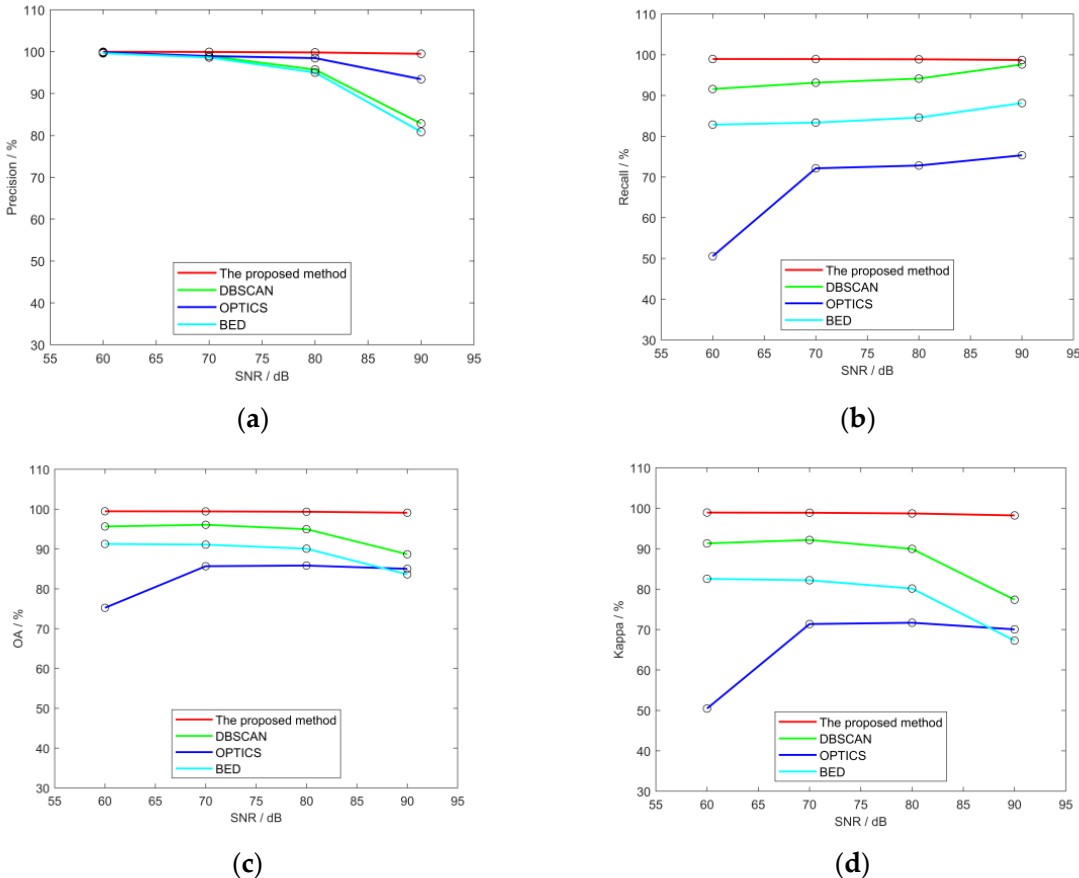

**Figure 13.** Curves of change in four validation indicators with SNR in experimental area A. (**a**) Precision; (**b**) Recall; (**c**) OA; (**d**) Kappa.

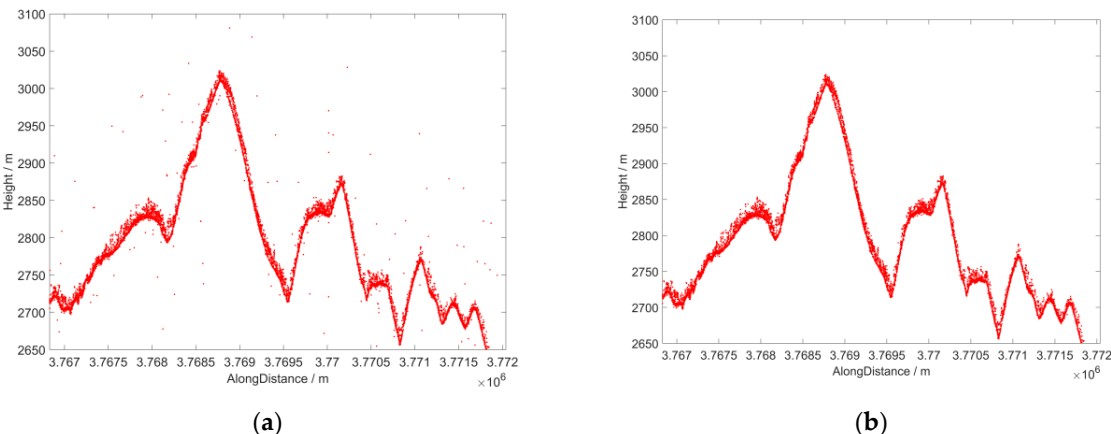

**Figure 14.** Overall distribution of photon data in experimental area B. (**a**) Original photon data; (**b**) signal photon data.

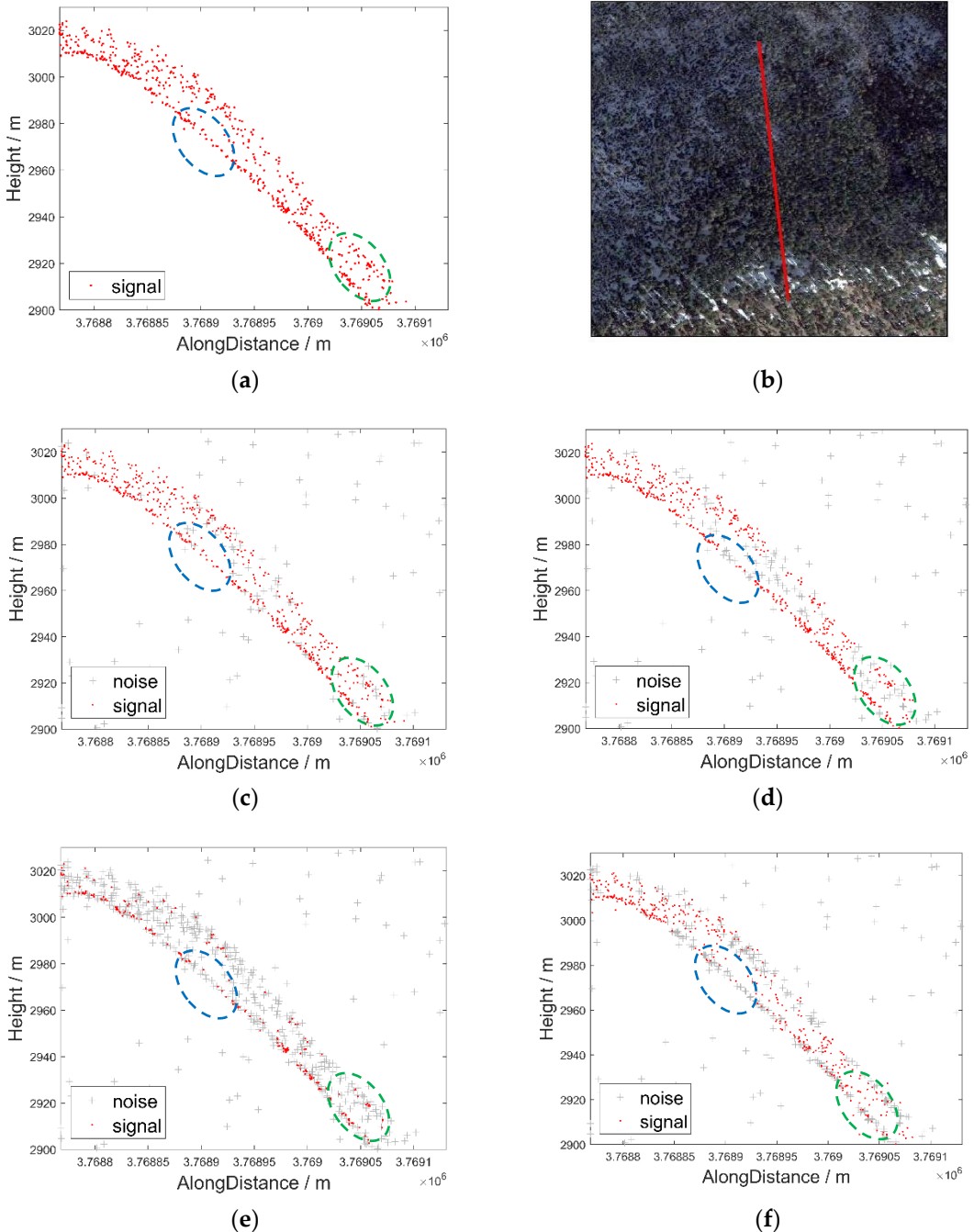

**Figure 15.** Comparison of the details of the results of the gt2R track in experimental area B (SNR = 70 dB). (**a**) Validation; (**b**) optical remote sensing image; (**c**) the proposed method; (**d**) DB-SCAN; (**e**) OPTICS; (**f**) BED.

Experimental area C is situated in a shallow marine region where ICESat-2 photon data exhibit two types of signal photons, namely water surface photons and water bottom photons. Due to the gradual propagation of the laser energy through the water, the signal photon density at the water surface is significantly higher than that at the water bottom (shown in Figure 17). As represented in Figure 18, both the DBSCAN and OPTICS algorithms solely identify water surface photons with high signal photon density (in the blue box), while disregarding water bottom signal photons with low photon density (in the green box). Moreover, some noise photons near the water surface are misclassified as signal photons (in the purple box), leading to a low recall result. Regarding the BED

algorithm, while it can extract most of the signal photons, it also misclassifies large number of noise photons as signal photons (in the purple box), leading to a low precision result. The proposed method effectively addresses these issues, resulting in a significant improvement in extraction performance and achieving the highest in all validation metrics. Similarly, Table 5 demonstrates the denoising performance of different denoising methods for different SNR levels in the area. As the SNR level increases, conventional methods omit more signal photons of the underwater photons, leading to progressively lower Recalls. But, the proposed method is a good solution to such a problem. As shown in the validation metrics, the proposed method achieves the highest overall accuracy of denoising results compared to other conventional methods. Figure 19 demonstrates the curves of the results of the four methods for all validation metrics under different SNRs in experimental area C. Compared to the conventional methods, the denoising results obtained by the proposed method (red line) are not affected by the different noise level, and the accuracy of the results are all better.

**Table 4.** Results of the gt2R track in experimental area B.

| | SNR (dB) | Precision | Recall | OA | Kappa |
|---|---|---|---|---|---|
| The proposed method | 60 | 99.98% | 97.63% | 98.81% | 97.62% |
| | 70 | 99.90% | 97.56% | 98.74% | 97.48% |
| | 80 | 99.70% | 97.48% | 98.60% | 97.20% |
| | 90 | 99.33% | 97.26% | 98.31% | 96.63% |
| DBSCAN | 60 | 98.51% | 92.95% | 95.82% | 91.64% |
| | 70 | 98.63% | 93.12% | 95.96% | 91.92% |
| | 80 | 94.61% | 96.25% | 95.44% | 90.87% |
| | 90 | 81.80% | 97.99% | 88.24% | 76.47% |
| OPTICS | 60 | 99.51% | 37.82% | 69.18% | 38.36% |
| | 70 | 98.56% | 57.42% | 78.54% | 57.08% |
| | 80 | 94.40% | 72.14% | 84.12% | 68.23% |
| | 90 | 89.07% | 78.56% | 84.64% | 69.28% |
| BED | 60 | 99.31% | 74.88% | 87.33% | 74.65% |
| | 70 | 97.72% | 75.55% | 87.05% | 74.09% |
| | 80 | 91.60% | 77.70% | 85.46% | 70.92% |
| | 90 | 74.16% | 83.17% | 77.37% | 54.73% |

**Table 5.** Results of the gt3L track in experimental area C.

| | SNR (dB) | Precision | Recall | OA | Kappa |
|---|---|---|---|---|---|
| The proposed method | 60 | 99.99% | 99.98% | 99.99% | 99.98% |
| | 70 | 99.99% | 99.98% | 99.99% | 99.97% |
| | 80 | 99.97% | 99.98% | 99.97% | 99.95% |
| | 90 | 99.96% | 99.89% | 99.93% | 99.85% |
| DBSCAN | 60 | 99.93% | 91.99% | 95.96% | 91.93% |
| | 70 | 99.93% | 91.99% | 95.96% | 91.93% |
| | 80 | 99.71% | 91.81% | 95.77% | 91.54% |
| | 90 | 99.71% | 91.81% | 95.77% | 91.54% |
| OPTICS | 60 | 99.95% | 87.12% | 93.53% | 87.07% |
| | 70 | 99.76% | 87.97% | 93.88% | 87.76% |
| | 80 | 99.20% | 88.64% | 93.96% | 87.93% |
| | 90 | 97.15% | 88.69% | 93.04% | 86.08% |
| BED | 60 | 99.72% | 99.78% | 99.75% | 99.50% |
| | 70 | 95.81% | 99.81% | 97.72% | 95.45% |
| | 80 | 79.49% | 99.83% | 87.03% | 74.07% |
| | 90 | 61.67% | 99.90% | 68.90% | 37.81% |

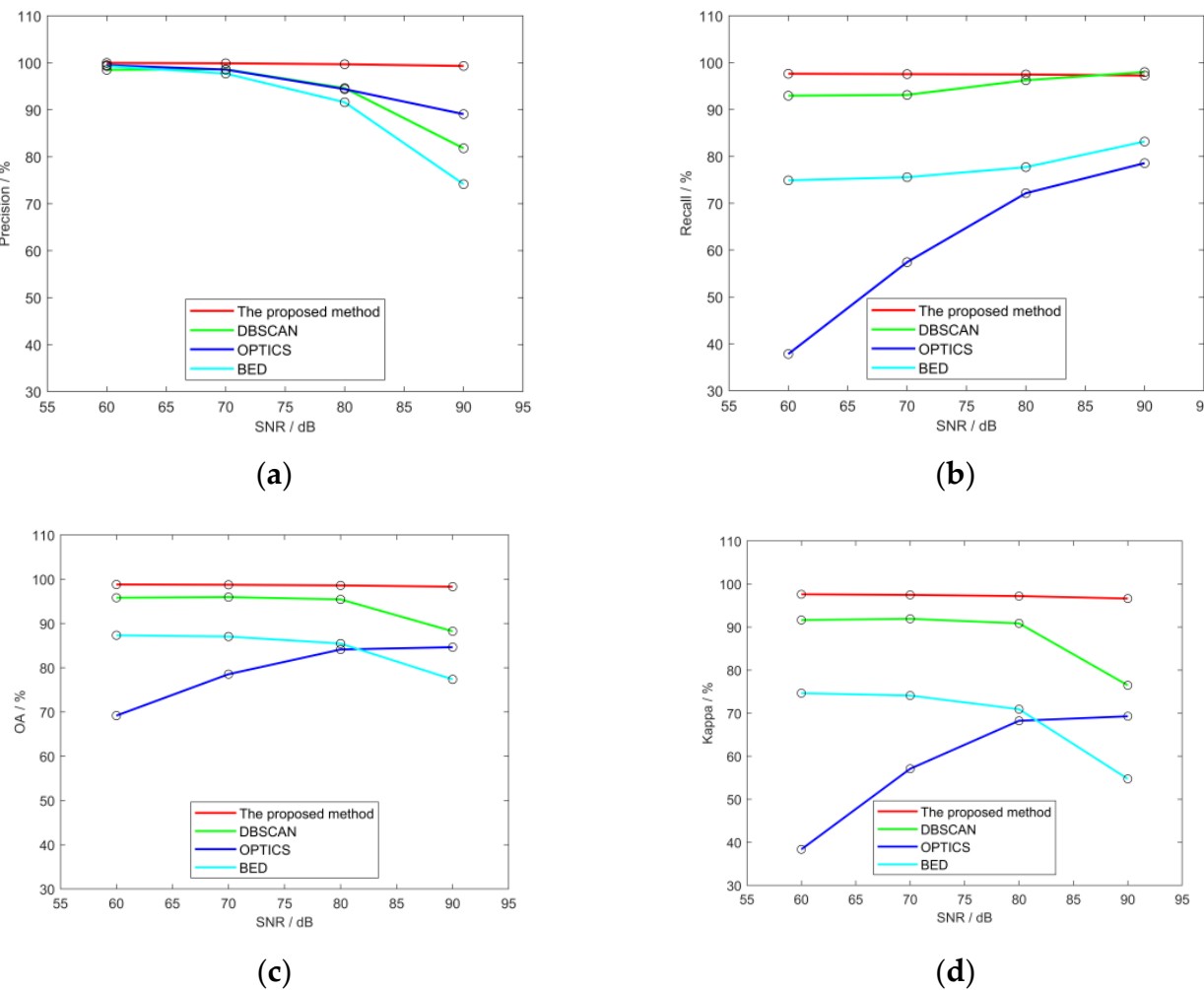

**Figure 16.** Curves of change in four validation indicators with SNR in experimental area B. (**a**) Precision; (**b**) Recall; (**c**) OA; (**d**) Kappa.

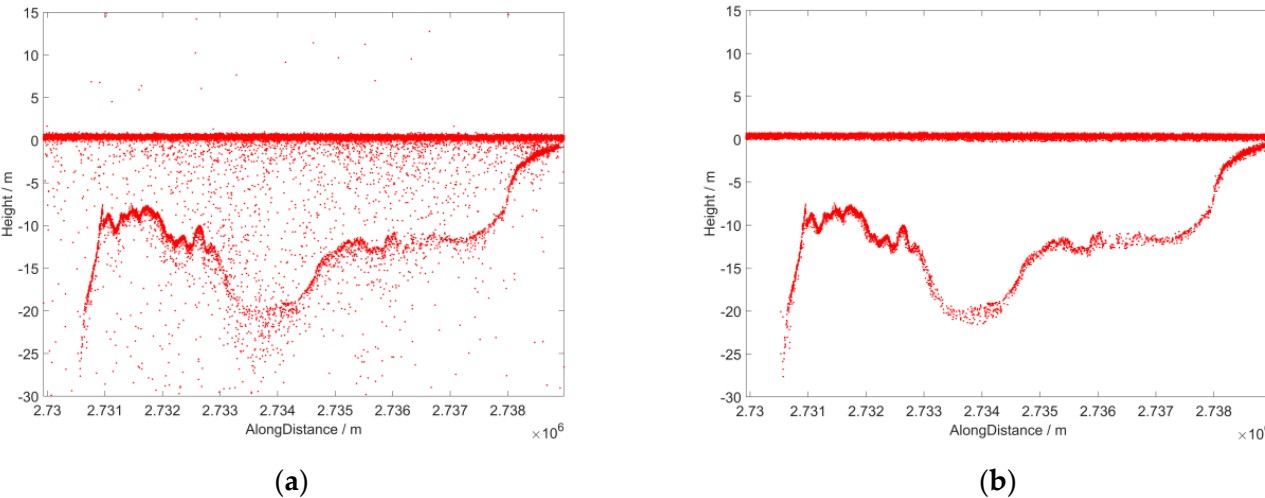

**Figure 17.** Overall distribution of photon data in experimental area C. (**a**) Original photon data; (**b**) signal photon data.

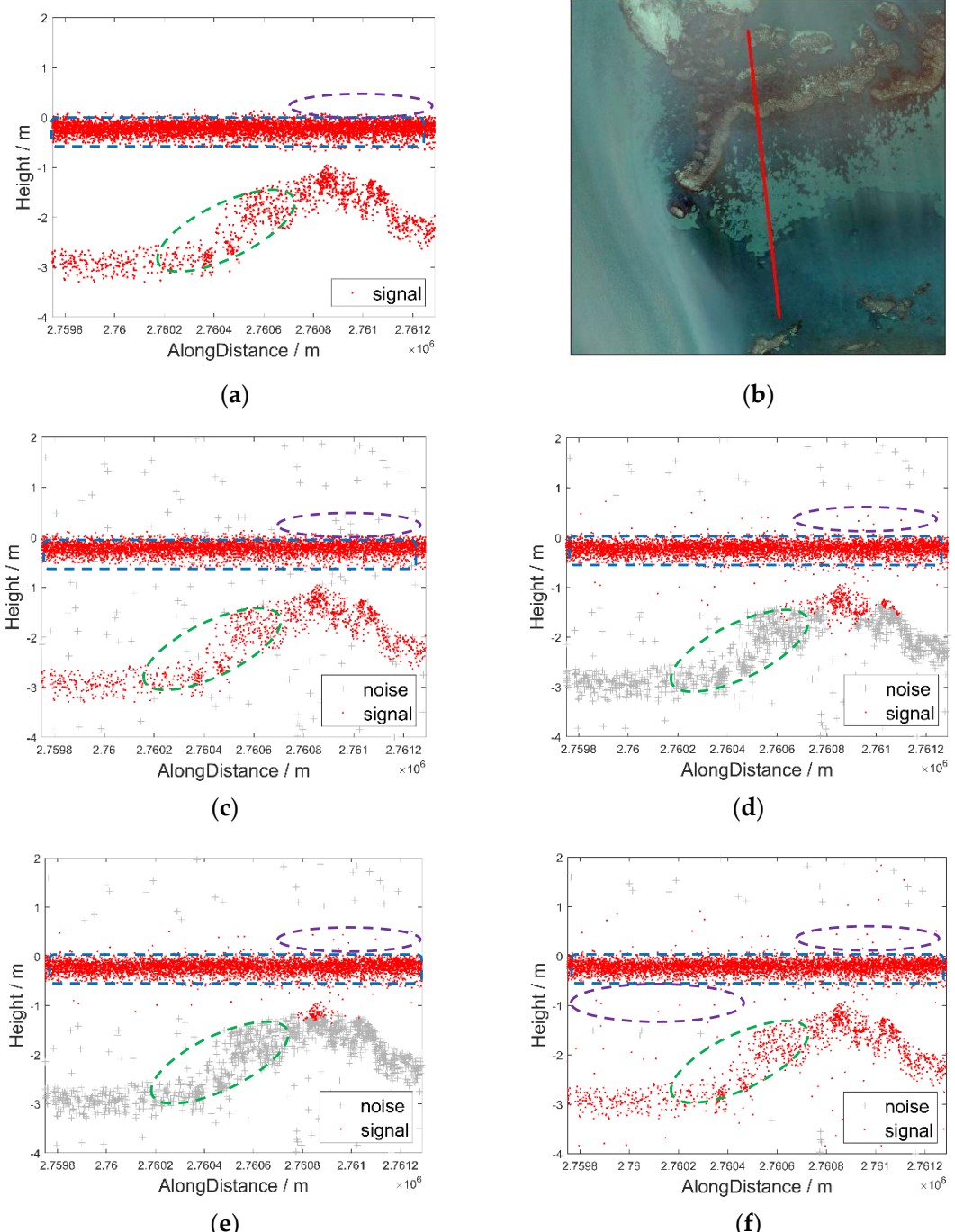

**Figure 18.** Comparison of the details of the gt3L track results in experimental area C (SNR = 80 dB). (**a**) Validation; (**b**) optical remote sensing image; (**c**) the proposed method; (**d**) DBSCAN; (**e**) OPTICS; (**f**) BED.

Experimental area D is located in a gently sloping mountainous area with a bare soil surface type. Figure 20 illustrates the real distribution of photon data in this area. Only one type of signal photon exists in the photon data of this area, which makes it an easy area to denoise, and the local density difference between signal photons and noise photons is significant. The main purpose of using these data is to compare the denoising level of different denoising methods in an uncomplicated area like flat bare soil. Figure 21 shows the details of the denoising results of different denoising methods. Table 6 demonstrates the accuracy metrics of the different denoising methods. It can be seen clearly that all

methods can achieve better denoising results in this area. Figure 22 demonstrates the curves of the results of the four methods for all validation metrics under different SNRs in experimental area D. Compared to the conventional methods, the proposed method is the most stable one.

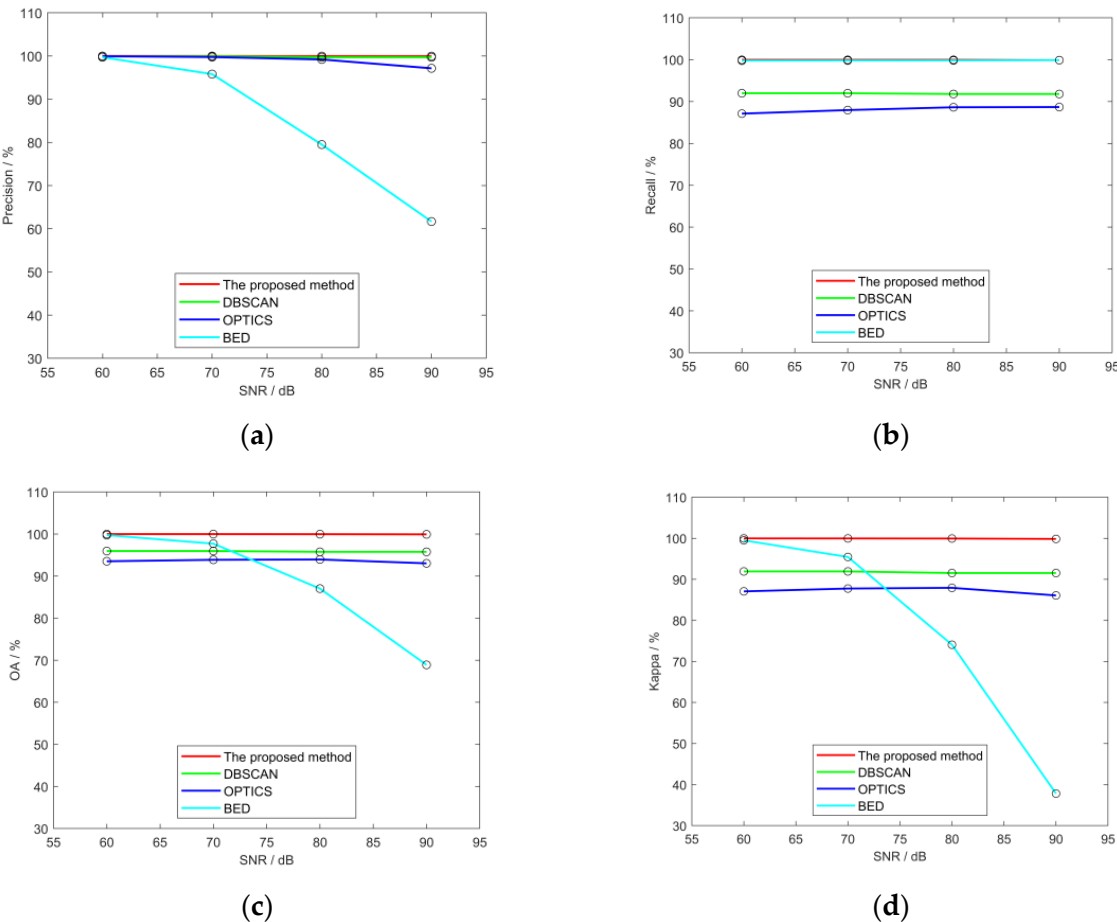

**Figure 19.** Curves of change in four validation indicators with SNR in experimental area C. (**a**) Precision; (**b**) Recall; (**c**) OA; (**d**) Kappa.

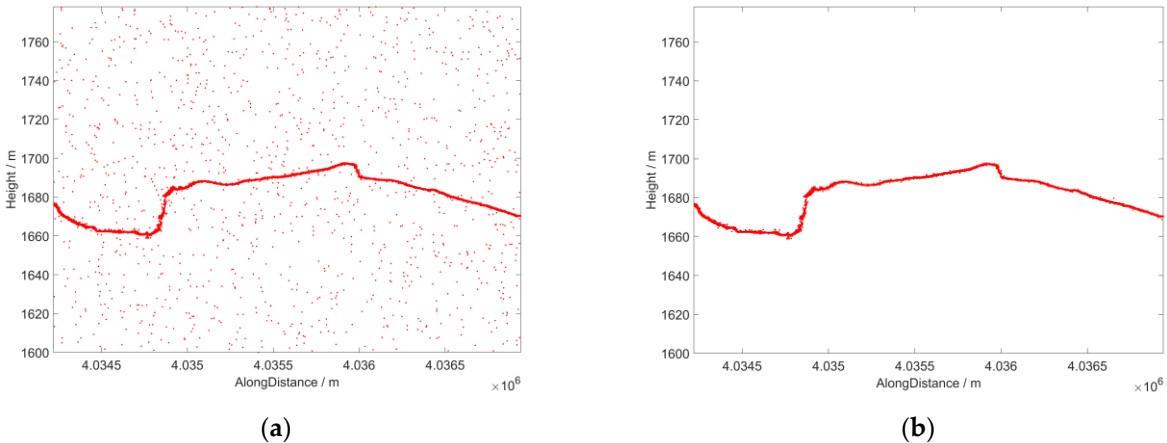

**Figure 20.** Overall distribution of photon data in experimental area D. (**a**) Original photon data; (**b**) signal photon data.

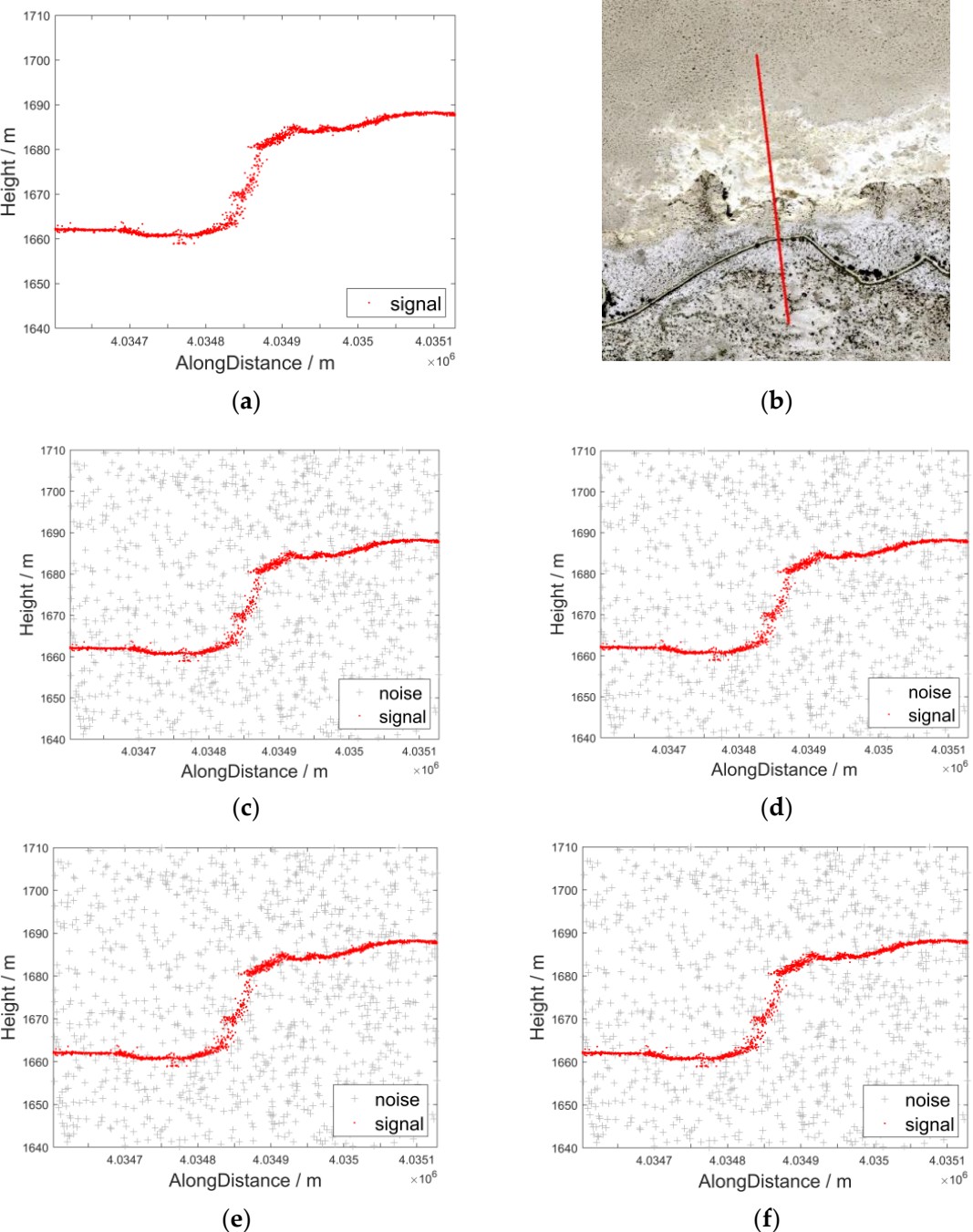

**Figure 21.** Comparison of the details of the gt2R track results in experimental area D (SNR = 80 dB). (**a**) Validation; (**b**) optical remote sensing image; (**c**) the proposed method; (**d**) DBSCAN; (**e**) OPTICS; (**f**) BED.

In summary, the proposed method outperforms the conventional methods in all simulation experiments, demonstrating the superiority of the proposed algorithm in terms of both local detail preservation and validation metrics. Additionally, the proposed method achieves high accuracy in three different types of areas and various SNR levels, highlighting its superior adaptability.

**Table 6.** Results of the gt2R track in the experimental area D.

|  | SNR (dB) | Precision | Recall | OA | Kappa |
|---|---|---|---|---|---|
| The proposed method | 60 | 99.89% | 99.84% | 99.87% | 99.73% |
|  | 70 | 99.97% | 99.84% | 99.91% | 99.81% |
|  | 80 | 99.67% | 99.82% | 99.75% | 99.49% |
|  | 90 | 99.68% | 99.81% | 99.75% | 99.49% |
| DBSCAN | 60 | 98.88% | 99.98% | 99.43% | 98.85% |
|  | 70 | 99.63% | 99.88% | 99.75% | 99.50% |
|  | 80 | 95.03% | 99.99% | 97.39% | 94.77% |
|  | 90 | 94.66% | 99.99% | 97.18% | 94.36% |
| OPTICS | 60 | 99.63% | 96.34% | 97.99% | 95.97% |
|  | 70 | 99.91% | 93.91% | 96.92% | 93.83% |
|  | 80 | 98.44% | 96.77% | 97.62% | 95.24% |
|  | 90 | 98.28% | 97.08% | 97.69% | 95.38% |
| BED | 60 | 96.53% | 99.60% | 98.01% | 96.01% |
|  | 70 | 99.54% | 99.50% | 99.52% | 99.05% |
|  | 80 | 97.22% | 98.91% | 98.04% | 96.09% |
|  | 90 | 97.07% | 98.75% | 97.88% | 95.77% |

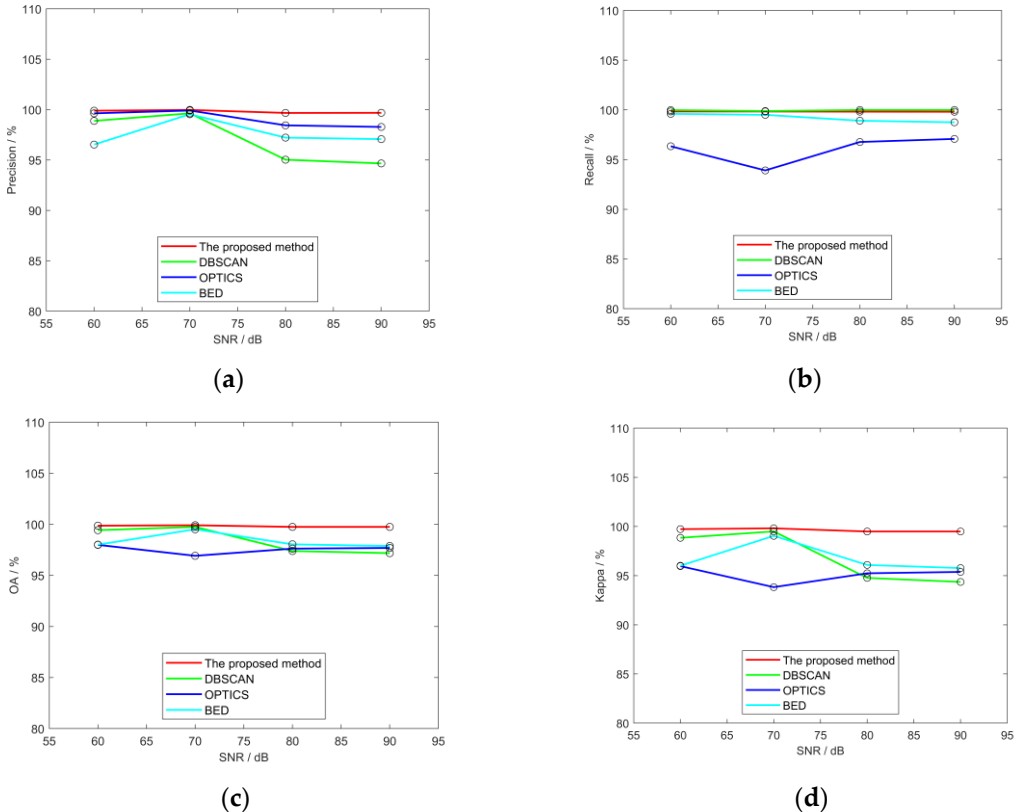

**Figure 22.** Curves of change in four validation indicators with SNR in experimental area D. (**a**) Precision; (**b**) Recall; (**c**) OA; (**d**) Kappa.

### 4.4. Real Reference Validation

To validate the accuracy of the signal photons extracted by proposed method, real validation reference data are gathered. Firstly, the raw ICESat-2 photon data from these four experimental areas are denoised using the proposed method, and then the signal photon elevation is extracted. Subsequently, the real photon elevation obtained from the topographic raster is compared with the signal photon elevation. In the third experimental

area which is located in shallow sea, refraction correction and tidal correction are conducted for the signal photons [41].

The results of the comparison in experimental area A are presented in Figure 23 and Table 7. Experimental area A is a mountainous area with an elevation range of 340 m. As depicted in Figure 23, most of the photon elevations accurately reflect the reference elevation, while only a few individual photons exhibit significant deviations, resulting in an RMSE of 3.40 m and an $R^2$ of 0.99. Table 7 presents the precision results of different denoising methods in each elevation interval. Lower MAE and MRE indicate that the signal photon elevations fit the real terrain elevations better and a lower RMSE indicates that the extracted signal photons contain fewer noise photons with large elevation errors. The proposed method achieves the best results in all three metrics compared to other conventional methods. These results also indicate that the signal photons extracted by the proposed method are more reliable.

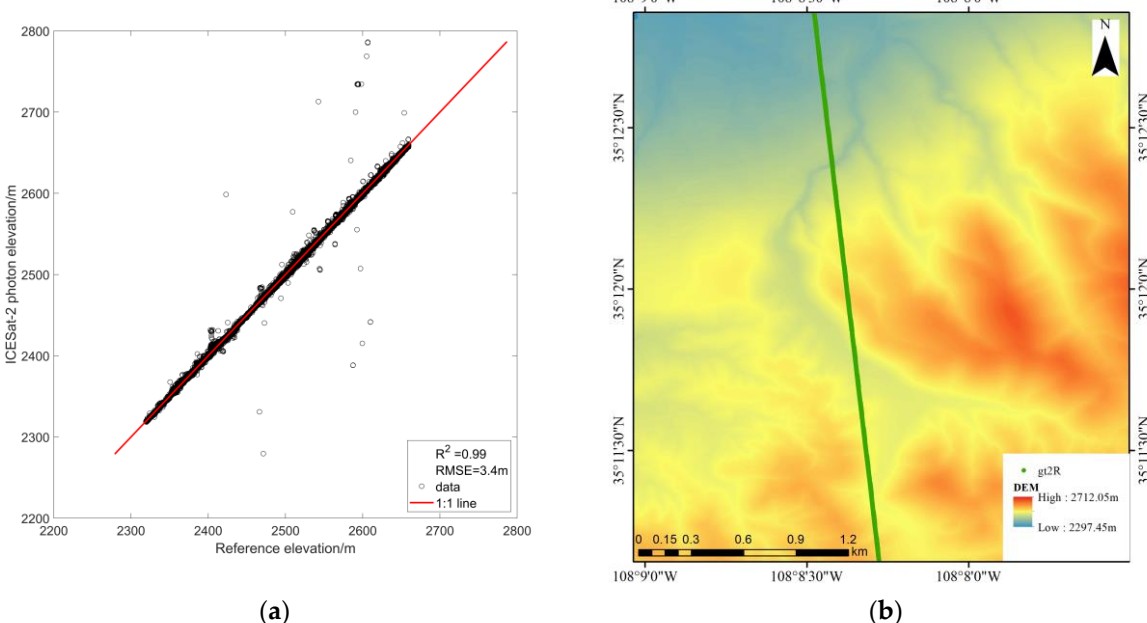

(**a**)  (**b**)

**Figure 23.** Schematic of real reference verification in experimental area A. (**a**) Scatterplot of elevation and actual elevation of gt2R track signal photons in experimental area A; (**b**) track distribution diagram in experimental area A.

**Table 7.** Accuracy results within different elevation intervals of the gt2R track in experimental area A (The best result for each evaluation interval is bolded).

|  | RMSE | MAE | MRE | Range |
|---|---|---|---|---|
| | 1.36 | 0.79 | 0.03% | 2319.22–2404.32 m |
| | 4.34 | 1.09 | 0.04% | 2404.32–2489.43 m |
| The proposed method | 1.93 | 0.79 | 0.03% | 2489.43–2574.54 m |
| | 4.12 | 0.64 | 0.02% | 2574.54–2659.64 m |
| | **3.40** | **0.75** | **0.03%** | Overall |
| | 3.64 | 1.10 | 0.05% | 2319.22–2404.32 m |
| | 3.09 | 1.26 | 0.05% | 2404.32–2489.43 m |
| DBSCAN | 4.25 | 1.09 | 0.04% | 2489.43–2574.54 m |
| | 3.34 | 0.89 | 0.03% | 2574.54–2659.64 m |
| | 3.58 | 1.09 | 0.04% | Overall |

**Table 7.** *Cont.*

|  | RMSE | MAE | MRE | Range |
|---|---|---|---|---|
| OPTICS | 3.93 | 0.75 | 0.03% | 2319.22–2404.32 m |
|  | 6.63 | 1.17 | 0.05% | 2404.32–2489.43 m |
|  | 4.18 | 0.88 | 0.03% | 2489.43–2574.54 m |
|  | 4.66 | 0.68 | 0.03% | 2574.54–2659.64 m |
|  | 4.85 | 0.87 | 0.03% | Overall |
| BED | 15.99 | 7.81 | 0.33% | 2319.22–2404.32 m |
|  | 22.85 | 11.36 | 0.46% | 2404.32–2489.43 m |
|  | 19.07 | 8.66 | 0.34% | 2489.43–2574.54 m |
|  | 20.38 | 12.34 | 0.47% | 2574.54–2659.64 m |
|  | 19.57 | 10.04 | 0.40% | Overall |

The comparison results between the denoised signal photon data and the real validation elevation in experimental area B are presented in Figure 24 and Table 8, respectively. Experimental area B is also mountainous terrain. As depicted in Figure 24, almost all of the photon elevations correspond well with the reference elevation, with only a few photons exhibiting significant differences.

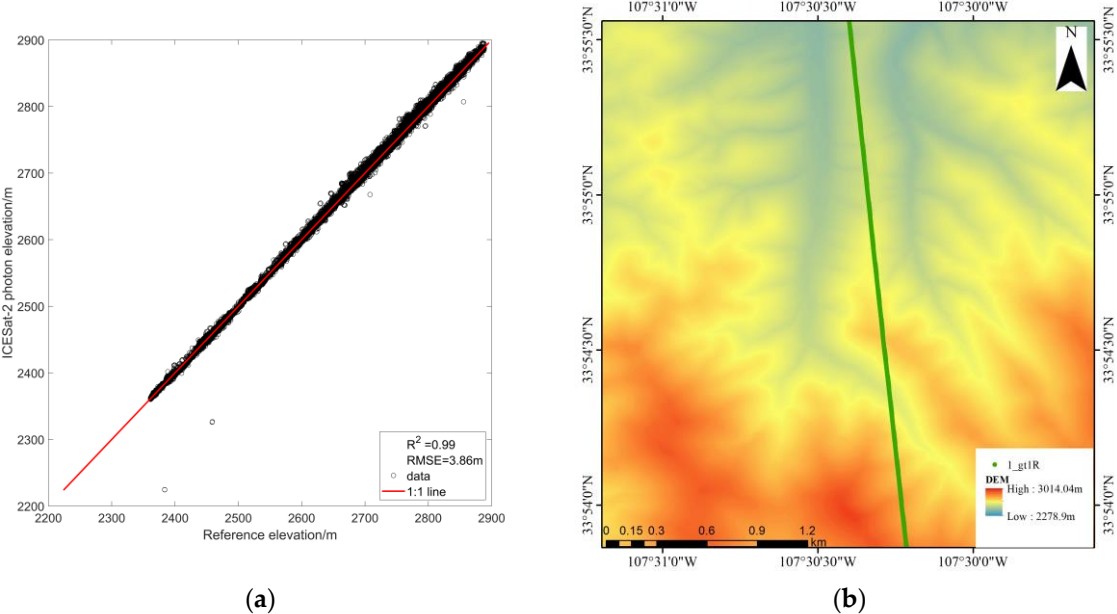

(**a**)  (**b**)

**Figure 24.** Schematic of real reference verification in experimental area B. (**a**) Scatterplot of elevation and actual elevation of gt1R track signal photons in the experimental area B; (**b**) track distribution diagram in the experimental area B.

The overall RMSE of experimental area B is 3.86 m, with an $R^2$ of 0.99, indicating the signal photon elevation is accurate with the reference elevation. Table 8 presents the precision results of different denoising methods in each elevation interval. Similar to experimental area A, the proposed method also achieves the best results in all three metrics compared to other conventional methods.

The comparison of the denoised signal photon data to the real validation elevation in experimental area C is listed in Figure 25 and Table 9. Experimental area C is located in a shallow sea area. Due to the complexity of this environment, including water quality conditions and turbidity affecting light propagation, the deepest water depth of 18 m resulted in an RMSE value of 0.81 m and an $R^2$ of 0.92. Table 8 presents the precision results of different denoising methods in each elevation interval. Both the DBSCAN method and the OPTICS method omit signal photons at water depths of 5–10 m. Similar to experimental

areas A and B, the proposed method also achieves the best results in all three metrics compared to other conventional methods.

**Table 8.** Accuracy results within different elevation intervals of the gt1R track in the experimental area B (The best result for each evaluation interval is bolded).

|  | RMSE | MAE | MRE | Range |
|---|---|---|---|---|
| The proposed method | 4.14 | 1.39 | 0.06% | 2361.07–2492.91 m |
|  | 2.31 | 1.67 | 0.07% | 2492.91–2624.75 m |
|  | 3.96 | 2.69 | 0.10% | 2624.75–2756.58 m |
|  | 4.40 | 3.09 | 0.11% | 2756.58–2888.42 m |
|  | **3.86** | **2.28** | **0.08%** | Overall |
| DBSCAN | 3.03 | 1.78 | 0.07% | 2361.07–2492.91 m |
|  | 4.22 | 2.59 | 0.10% | 2492.91–2624.75 m |
|  | 6.28 | 4.28 | 0.16% | 2624.75–2756.58 m |
|  | 6.38 | 4.75 | 0.17% | 2756.58–2888.42 m |
|  | 4.98 | 3.35 | 0.13% | Overall |
| OPTICS | 10.38 | 2.27 | 0.09% | 2361.07–2492.91 m |
|  | 12.07 | 3.07 | 0.12% | 2492.91–2624.75 m |
|  | 10.46 | 3.76 | 0.14% | 2624.75–2756.58 m |
|  | 9.18 | 4.19 | 0.15% | 2756.58–2888.42 m |
|  | 10.52 | 3.32 | 0.13% | Overall |
| BED | 15.63 | 3.76 | 0.16% | 2361.07–2492.91 m |
|  | 16.69 | 4.82 | 0.19% | 2492.91–2624.75 m |
|  | 14.15 | 5.48 | 0.20% | 2624.75–2756.58 m |
|  | 14.32 | 6.04 | 0.21% | 2756.58–2888.42 m |
|  | 15.20 | 5.02 | 0.19% | Overall |

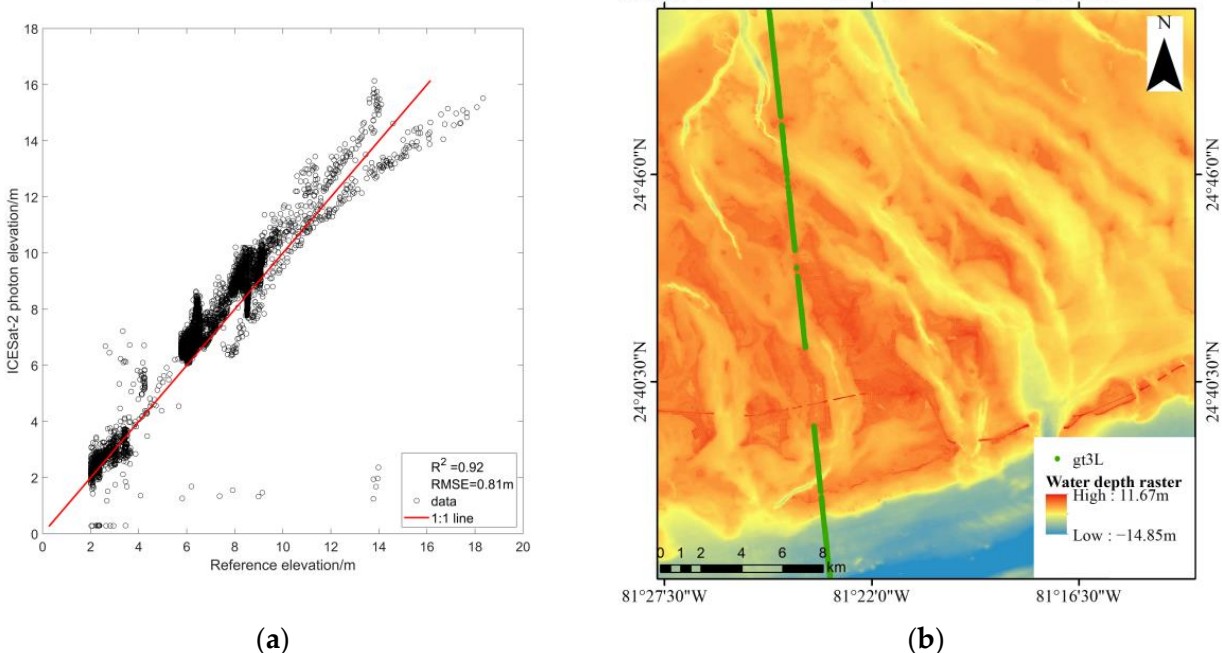

(**a**)  (**b**)

**Figure 25.** Schematic of real reference verification in experimental area C. (**a**) Scatterplot of elevation and actual elevation of gt3L track signal photons in experimental area C; (**b**) track distribution diagram in experimental area C.

**Table 9.** Accuracy results within different elevation intervals of the gt3L track in experimental area C (The best result for each evaluation interval is bolded).

|                     | RMSE | MAE | MRE | Range |
|---------------------|------|------|--------|---------|
| The proposed method | 1.11 | 0.44 | 13.25% | 2–5 m |
|                     | 0.55 | 0.71 | 10.37% | 5–10 m |
|                     | 0.90 | 0.94 | 8.51%  | 10–15 m |
|                     | 1.46 | 1.74 | 11.84% | 15–20 m |
|                     | 0.81 | **0.69** | **10.74%** | Overall |
| DBSCAN              | 0.64 | 0.77 | 26.69% | 2–5 m |
|                     | 0.71 | 0.76 | 10.77% | 5–10 m |
|                     | 0.61 | 1.43 | 16.70% | 10–15 m |
|                     | -    | -    | -      | 15–20 m |
|                     | 0.65 | 0.99 | 18.05% | Overall |
| OPTICS              | 0.44 | 0.77 | 29.73% | 2–5 m |
|                     | 0.50 | 0.73 | 10.67% | 5–10 m |
|                     | 0.29 | 1.06 | 12.81% | 10–15 m |
|                     | -    | -    | -      | 15–20 m |
|                     | 0.41 | 0.85 | 17.74% | Overall |
| BED                 | 2.44 | 1.42 | 29.32% | 2–5 m |
|                     | 0.72 | 0.75 | 10.97% | 5–10 m |
|                     | 1.08 | 1.11 | 10.69% | 10–15 m |
|                     | 0.52 | 2.11 | 15.28% | 15–20 m |
|                     | 1.19 | 1.35 | 16.57% | Overall |

The results of the comparison in experimental area D are presented in Figure 26 and Table 10. Since the resolution of the validated DEM in experimental area D is 30 m, which is much lower compared to other experimental areas, the description of the terrain is much coarser. In contrast, the elevations obtained by the signal photons are much finer. The difference between them results in a poorer fit between the signal photon elevation and the actual terrain elevation than that in the previous three experimental areas, resulting in an RMSE of 4.53 m and an $R^2$ of 0.87. However, there were still many signal photons that fit the terrain. Table 10 presents the precision results of different denoising methods in each elevation interval. Since the denoising effect of the four compared methods is about the same in this area, there is no distinction between the four in the accuracy results.

**Table 10.** Accuracy results within different elevation intervals of the gt2R track in experimental area D (The best result for each evaluation interval is bolded).

|                     | RMSE/m | MAE/m | MRE | Range |
|---------------------|--------|-------|-------|-------------|
| The proposed method | 3.00 | 2.71 | 0.16% | 1659–1672 m |
|                     | 2.96 | 3.40 | 0.20% | 1672–1685 m |
|                     | 2.95 | 2.80 | 0.17% | 1685–1698 m |
|                     | 3.17 | 9.94 | 0.58% | 1698–1711 m |
|                     | 3.02 | **4.71** | **0.28%** | Overall |
| DBSCAN              | 2.95 | 2.71 | 0.16% | 1659–1672 m |
|                     | 2.98 | 3.45 | 0.21% | 1672–1685 m |
|                     | 2.96 | 2.81 | 0.17% | 1685–1698 m |
|                     | 3.16 | 9.92 | 0.58% | 1698–1711 m |
|                     | **3.01** | 4.72 | **0.28%** | Overall |
| OPTICS              | 2.79 | 2.61 | 0.16% | 1659–1672 m |
|                     | 2.99 | 3.30 | 0.20% | 1672–1685 m |
|                     | 2.93 | 2.79 | 0.17% | 1685–1698 m |
|                     | 3.21 | 10.27 | 0.60% | 1698–1711 m |
|                     | 2.98 | 4.75 | **0.28%** | Overall |

**Table 10.** *Cont.*

|  | RMSE/m | MAE/m | MRE | Range |
|---|---|---|---|---|
|  | 3.09 | 2.75 | 0.17% | 1659–1672 m |
|  | 2.97 | 3.42 | 0.20% | 1672–1685 m |
| BED | 2.95 | 2.80 | 0.17% | 1685–1698 m |
|  | 3.16 | 9.93 | 0.58% | 1698–1711 m |
|  | 3.04 | 4.73 | **0.28%** | Overall |

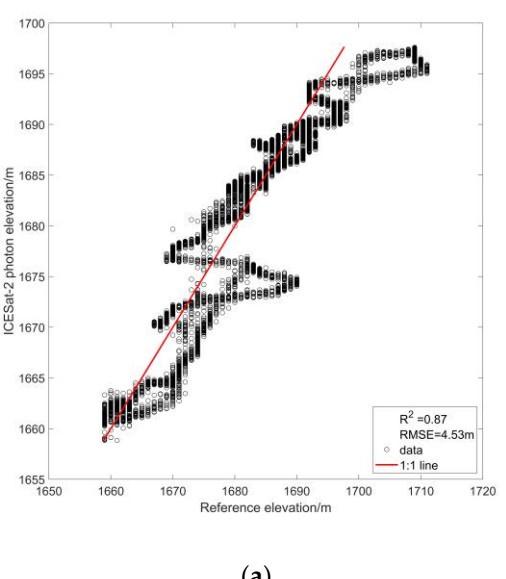

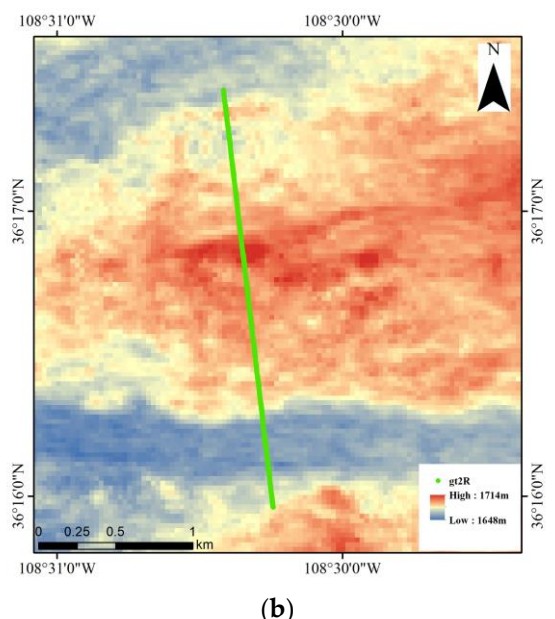

(**a**)  (**b**)

**Figure 26.** Schematic of real reference verification in experimental area D. (**a**) Scatterplot of elevation and actual elevation of gt2R track signal photons in experimental area D; (**b**) track distribution diagram in experimental area D.

In summary, the signal photon elevation extracted by the proposed method is consistent with the real validation elevation, indicating the correctness of the proposed method.

## 5. Discussion

### 5.1. Inter-CNN Model Comparison

In this section, VGG16, ResNet18, ResNet50, GoogLeNet, and DenseNet201 are selected for comparison, and the performance of the five network models is demonstrated using the datasets from experimental areas A and B, respectively. Finally, all the data results from the two experimental areas are averaged by the SNR presented in Table 11.

**Table 11.** Comparison of results between different CNN models (The best result for each evaluation interval is bolded).

| Name | SNR (dB) | Precision | Recall | OA | Kappa |
|---|---|---|---|---|---|
|  | 60 | 99.66% | 97.67% | 98.67% | 97.35% |
|  | 70 | 99.15% | 97.00% | 98.09% | 96.18% |
| DenseNet201 | 80 | 98.99% | 96.66% | 97.84% | 95.67% |
|  | 90 | 98.33% | 92.62% | 95.54% | 91.09% |
|  | Average | 99.03% | 95.99% | 97.54% | 95.07% |

**Table 11.** *Cont.*

| Name | SNR (dB) | Precision | Recall | OA | Kappa |
|---|---|---|---|---|---|
| GoogLeNet | 60 | 99.72% | 98.45% | 99.09% | 98.18% |
| | 70 | 99.27% | 98.37% | 98.83% | 97.66% |
| | 80 | 98.87% | 98.48% | 98.68% | 97.36% |
| | 90 | 97.90% | 97.96% | 97.93% | 95.87% |
| | Average | 98.94% | **98.32%** | **98.63%** | **97.27%** |
| ResNet18 | 60 | 99.50% | 97.56% | 98.54% | 97.07% |
| | 70 | 98.72% | 97.03% | 97.89% | 95.78% |
| | 80 | 98.24% | 96.93% | 97.60% | 95.20% |
| | 90 | 97.35% | 94.57% | 96.00% | 92.01% |
| | Average | 98.45% | 96.52% | 97.51% | 95.02% |
| ResNet50 | 60 | 99.94% | 94.87% | 97.41% | 94.82% |
| | 70 | 99.86% | 93.57% | 96.72% | 93.45% |
| | 80 | 99.77% | 93.86% | 96.83% | 93.67% |
| | 90 | 99.65% | 87.21% | 93.47% | 86.95% |
| | Average | **99.80%** | 92.38% | 96.11% | 92.22% |
| VGG16 | 60 | 99.69% | 98.43% | 99.06% | 98.13% |
| | 70 | 99.23% | 98.05% | 98.65% | 97.30% |
| | 80 | 98.72% | 98.12% | 98.43% | 96.85% |
| | 90 | 97.98% | 95.52% | 96.79% | 93.57% |
| | Average | 98.90% | 97.53% | 98.23% | 96.46% |

From Table 11, it can be observed that GoogleNet is able to extract signal photons most accurately in terms of the overall average Recall, Overall Accuracy, and Kappa coefficient. Only ResNet50 has the highest accuracy metric, but the difference between GoogLeNet and ResNet50 is negligible. Therefore, it is reasonable to utilize the GoogLeNet as the backbone model.

*5.2. Attentional Mechanisms*

In the proposed method, the attention mechanism module of CBAM is incorporated into the GoogLeNet model. Since the portion with data in the photon images is not very large, most of the spaces are blank areas without data. With the attention mechanism, the proposed method can focus on learning the crucial features more and improve the discrimination between signal and noise. The network classification results before and after using CBAM are compared in this article. The datasets of experimental areas A and B are used to train and test these two network models separately, and the results are presented in Table 12. In terms of the accuracy of the results, after using the attention mechanism, the accuracy of the results has been improved to some extent at each noise level. The higher the noise level, the greater the improvement. In terms of the average results, the accuracy of each validation metric is improved after adding the attention mechanism, which indicates that the attention mechanism has positive significance for the denoising of photon images in this article.

**Table 12.** Comparison of the accuracy of results before and after using CBAM (The best result for each evaluation interval is bolded).

| Name | SNR (dB) | Precision | Recall | OA | Kappa |
|---|---|---|---|---|---|
| GoogLeNet | 60 | 99.72% | 98.45% | 99.09% | 98.18% |
| | 70 | 99.27% | 98.37% | 98.83% | 97.66% |
| | 80 | 98.87% | 98.48% | 98.68% | 97.36% |
| | 90 | 97.90% | 97.96% | 97.93% | 95.87% |
| | Average | 98.94% | 98.32% | 98.63% | 97.27% |
| GoogLeNet + CBAM | 60 | 99.94% | 98.75% | 99.34% | 98.69% |
| | 70 | 99.86% | 98.69% | 99.28% | 98.56% |
| | 80 | 99.67% | 98.58% | 99.13% | 98.26% |
| | 90 | 99.43% | 98.40% | 98.92% | 97.84% |
| | Average | **99.72%** | **98.60%** | **99.17%** | **98.33%** |

## 6. Conclusions

In this article, to address the problems of conventional denoising algorithms, a novel CNN-based ICESat-2 signal photon extraction method is proposed. This method transforms the photon and its neighborhood into a 2D image to preserve the position and shape features. Then, GoogLeNet fused with CBAM is utilized to learn photon semantic features, which can simultaneously extract signal photons with different photon densities to avoid misclassification of noise photons and the CBAM enhances the ability of network to focus more on learning the crucial features and improves its discriminative ability. In the experiments, the simulation data with different SNR and real validation data are presented to validate the proposed method. From the results, the denoising results obtained by the proposed method in all four simulation experiments are much more accurate than the conventional methods in all the validation metrics and the OA is above 98%, which demonstrates the superiority of the proposed method. In the real validation experiments, the RMSE values are 3.40 m and 3.86 m in the mountainous area with altitude spans of 340 m and 527 m, respectively, and the RMSE value is 0.81 m in the shallow sea area with a maximum water depth of 18 m. In the mountainous area, the RMSE value is 4.53 m and the $R^2$ values are above 0.87 in all experimental areas, indicating that the elevation obtained from the signal photons closely approximates the reference validation data, which can prove the correctness of the signal photons obtained by the proposed method. In the Section 5, this article compares the accuracy of the results between different CNN models and evaluates the impact of CBAM on model performance. The results of the proposed method are the most accurate.

**Author Contributions:** Conceptualization, W.Q. and Y.S.; methodology, W.Q. and Y.S.; software, W.Q.; validation, W.Q. and Y.S.; formal analysis, Y.S. and H.G.; investigation, H.G.; resources, Y.Z. and H.Z.; data curation, W.Q.; writing—original draft preparation, W.Q.; writing—review and editing, Y.S. All authors have read and agreed to the published version of the manuscript.

**Funding:** This research was funded by the Mangrove monitoring and change factor analysis based on multi-source satellite remote sensing data (grant no. 202301001), and by the Integration and Application Demonstration in the Marine Field (grant no. 2020010004); and by the Automated Identifying of Environment Changes Using Satellite Time-Series, Dragon 5 Cooperation 2020–2024 (grant no. 57971).

**Data Availability Statement:** Not applicable.

**Acknowledgments:** We would like to express appreciation to the National Aeronautics and Space Administration (NASA) for providing the ICESat-2 data used in the article. We also appreciate the National Oceanic and Atmospheric Administration (NOAA) and G-LiHT science team for providing

the Real Validation Data used in this article. Moreover, we thank the anonymous reviewers and members of the editorial team for their constructive comments.

**Conflicts of Interest:** The authors declare no conflict of interest.

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
