# Peer review of "A Novel ICESat-2 Signal Photon Extraction Method Based on Convolutional Neural Network"

_remotesensing, doi:10.3390/rs16010203_

Round 1

Reviewer 1 Report

Comments and Suggestions for Authors

The paper provides a clear and logically structured overview of the main topics covered. It thoroughly analyzes and discusses the specific methods of using GoogLeNet and CBAM for photon signal extraction. The writing is rigorous and takes into account the heterogeneity of the regions, conducting multiple samplings of the region data. The experimental results are verified for accuracy using multiple methods, making the conclusions reliable. However, the paper does not explain the labels of the experimental data, so it is suggested to add relevant content in this regard. In summary, the article has a tight logical structure and promotes the application of GoogLeNet and CBAM in the field of photon signal extraction, providing valuable reference for others. Comments on the Quality of English Language

The English is good enough for me, only minor modifications are needed.

Author Response

Dear reviewer:

We gratefully thank you for your time spend making your constructive remarks and useful suggestions, which has significantly raised the quality of the manuscript and has enables us to improve the manuscript. We have studied your comments carefully and modified the manuscript. Revised portion are highlighted in the paper. The responses to your comments are as following:

  1. Comment: However, the paper does not explain the labels of the experimental data, so it is suggested to add relevant content in this regard.

Response: Thank you for your suggestion very much. In the paper, we have added a section "2.3 Training dataset" to explain the construction and labeling process of the training dataset. In the experimental process, the simulation data are used to train the network model. The signal photons are manually labeled based on reference data such as terrain, while the noise photons are Gaussian white noise of different levels added to the signal photons.

Reviewer 2 Report

Comments and Suggestions for Authors

The study introduces a new method developed for processing ICESat-2 photon data and emphasizes that this method performs better than traditional methods and has been successfully verified with both simulation and real data.

1- According to which results were the parameters given in table 2 determined in the selection of hyperparameters? can the result of the tested parameters be shared in the study?

2- Why has the proposed model not been tried with other satellite images?

3- Why was a comparison with other reference articles not given?

Author Response

Dear reviewer:

We gratefully thank you for your time spend making your constructive remarks and useful suggestions, which has significantly raised the quality of the manuscript and has enables us to improve the manuscript. We have studied your comments carefully and modified the manuscript. Revised portion are highlighted in the paper. The responses to your comments are as following:

  1. Comment: 1- According to which results were the parameters given in table 2 determined in the selection of hyperparameters? can the result of the tested parameters be shared in the study?

Response: The hyperparameter settings during training are shown in Table 2. These hyperparameters were set based on the publicly available code projects of the authors of GoogLeNet. These hyperparameters were subsequently fine-tuned during the experiments based on the accuracy results of the simulations. We finally decided to use these parameters after we felt that a satisfactory accuracy was achieved. Hyperparameters can be shared during testing. The hyperparameters we used are the same as the training hyperparameters, so we did not list these parameters additionally.

  1. Comment: 2- Why has the proposed model not been tried with other satellite images?

Response: For optical or SAR remote sensing images, since the data form of ICESat-2 photon data is completely different from them, and optical and SAR remote sensing images do not need such a denoising process like ICESat-2 photon data. So we did not apply the proposed model to these data. As for other satellite lidar data such as GDEI, since GDEI is waveform data, the data format is also different from ICESat-2. Thus the proposed model cannot be directly applied. In the future we will also actively expand the application scenarios of the proposed model.

  1. Comment: 3- Why was a comparison with other reference articles not given?

Response: The conventional methods used for comparison in this paper are actually from the reference in the introduction, where the OPTICS method is from reference [26], the BED method is from reference [24], and the DNSCAN method is from reference [20]. Among them, the DBSCAN algorithm, due to its wider application ground, we use a generalized modification, i.e., modifying the circular domain to an elliptical domain. We have included references to these three methods in Line310-312 to allow the reader to have a clearer understanding of where the compared methods come from.

Reviewer 3 Report

Comments and Suggestions for Authors

The authors have certainly written an interesting work. The results of the work are of high practical importance.

However, I have a few questions and comments.

1. Neural network training is based on simulation results. Comparison with other methods is also based on the results of modeling. But the real result will be only if, after the proposed processing method, the height of the relief is determined more accurately. However, the authors simply state the fact that their method roughly hits the right point, there is no comparison with other methods here. It turns out that the authors demonstrate the advantage of the method only on model data!

2.  The selected scan areas for the experiment are unsuccessful. In my opinion, the most convenient areas for initial experimentation are areas of the surface without vegetation and with the most even relief. To do this, you can use both natural objects and artificial objects.

3. In my opinion, the authors should add one more track to the three experimental tracks available on a flat section of the earth's surface and compare it with other methods precisely in terms of the accuracy of determining the height.

Author Response

Dear reviewer:

We gratefully thank you for your time spend making your constructive remarks and useful suggestions, which has significantly raised the quality of the manuscript and has enables us to improve the manuscript. We have studied your comments carefully and modified the manuscript. Revised portion are highlighted in the paper. The responses to your comments are as following:

  1. Comment: 1. Neural network training is based on simulation results. Comparison with other methods is also based on the results of modeling. But the real result will be only if, after the proposed processing method, the height of the relief is determined more accurately. However, the authors simply state the fact that their method roughly hits the right point, there is no comparison with other methods here. It turns out that the authors demonstrate the advantage of the method only on model data!

Response: Thank you for your suggestion, we did miss something in this aspect. We have now added comparisons of traditional methods in each of the real validation experiments, and these results are put into Tables 7, 8, 9, and 10. Ultimately, in terms of results, the method in this paper has the same highest accuracy in the real validation in each experimental area, and further demonstrates the advantages of the method in this paper.

  1. Comment: 2. The selected scan areas for the experiment are unsuccessful. In my opinion, the most convenient areas for initial experimentation are areas of the surface without vegetation and with the most even relief. To do this, you can use both natural objects and artificial objects.
  2. In my opinion, the authors should add one more track to the three experimental tracks available on a flat section of the earth's surface and compare it with other methods precisely in terms of the accuracy of determining the height.

Response: Thank you for your suggestion gratefully. In addition to the original three experimental areas, we have added one more experimental area D, which is located in the flat mountainous area. And we have increased the number of simulation and real validation experiments for experimental area D. The surface type of this experimental area is bare soil with no vegetation cover. The signal photons in this area are of a single class and differ from the noise photons. The results of the final simulation experiments with real verification experiments show that the methods in this paper do not have a significant difference from the conventional methods in such a region. All four methods selected in this paper can get better results. This set of experiments also shows that the methods in this paper can still achieve better results in simple scenarios.

Reviewer 4 Report

Comments and Suggestions for Authors

Qin et al. applied a deep learning model to better denoise photon data measured by the ICESat-2/ATLAS instrument. The proposed idea looks reasonable to me. However, the manuscript has to be better organized and the writing needs to be improved. Specifically, a lot of the details about the DL model and the training process are not clearly clarified in an organized manner, which are essential for reproduction of the results. These details about the methodology have to be clarified before acceptance for publication of the paper.

Specific comments:

Abstract: A large part of the abstract should be move to introduction. For example, L20-22 is too deep into details about the model, which should not appear in abstract.

L17: Is the term "semantic information" appropriately used in this application?

L57-59: It is hard to follow this sentence, revise?

L61-76: a lot of proposed methods are mentioned here. However, I am afraid that the readers won't appreciate this section because all these mentions are too brief. For example, what advantages and disadvantages these methods have? Which ones have an overall better performance? These discussions form the foundation of the motivation of this work.

L77: The authors jump directly from ICESat-2 photon denoising into 3D point cloud data denoising, of which the transition is kind of abrupt for the reader. The linkage between the two topics has to be better clarified.

L109-110: reference?

L126-128, also Figure 1: Experimental areas A and B seem to be very similar to each other. I suspect that would lead to the overfitting of your model towards this surface type.

L135-137: reference needed.

Table 1: table 1 is currently hard to read. I would recommend specifically include dates and times of satellite overpasses in the table.

L143-145: DTM and G-LiHT are not defined in the manuscript.

Figure 1: you have a figure here showing the location of the 3 experimental areas and the 9 tracks on map. But this figure is not mentioned in the main text.

L151: this link is not working.

Figure 2(2): it seems like the "image" data sets were generated by randomly sampling the whole ICESat-2 image. This should be clearly stated in the main text, otherwise it would be difficult to reproduce the results in this paper.

Figure 2(3): I can understand FC stands for fully-connected layers and what max pooling is. But the authors should not assume that the general audience are familiar with these terms.

Figure 2(4): what is "absolute verification"? Do you mean the validation of the elevation estimated by the proposed DL-based method using the independent elevation data sets mentioned in Section 2.2?

Figure 3: What is the output of your DL model? Is it a binary output or 2D images? I don't think this schematic diagram is correct.

L200-201: The three channels seem to have very different magnitudes. Are any scaling or additional transformation functions applied on these input channels?

L209: references needed here.

L222-231: The authors also need to show a diagram similar to Figs. 3 and 4 for the attention module.

L242-244: These two sentences should be one sentence.

L246-247: So does it mean that the ground truth is human-labelled? This information should be made clear in the Section 2.

L250-251: hard to follow this sentence

The authors kept using the terms "simulation experiment" and "experimental process", but it turns out to me that they are referring to the training process of the DL model. Please clarify this.

L263-265: What does each "piece of data" mean here? Is it referring to the different tracks?

L268-270: If a separate DL model is needed for each surface type, I would say this method is much less useful than a universal DL model that is able to deal with multiple surface types. I also still don't understand why the authors chose to include two areas with almost the same surface features. Is it possible to train the DL model using the combined data set of the three experimental areas?

Table 2: batch size of 2 is too small and could potentially lead to overfitting.

It is surprising to me that it is not mentioned in the manuscript what loss function was used in the training process. If the DL model has binary output format for the classification problem, I'd guess something like cross-entropy was being used here. It is mentioned in L257 that MAE and MRE were used. But what is the exact format of the loss function, as shown by the curves in Figs. 5(b) and 6(b)?

From Figs. 5 and 6, I think overfitting occurred in the training process. It would be great to mention if any efforts are made to avoid overfitting. Also these two figures are displayed but not discussed in the text.

Figure 7: What is the reasoning behind the design of the size of the window for feature selection?

Figure 10(a): Is this validation data set independent from the training process?

Tables 3, 4 and 5: I think plotting these numbers as curves as functions of SNR would be very helpful to show the robustness of different methods against noise level.

Figures 15 and 16: It would be interesting to see the estimated canopy heights, and (possibly) comparison against some independent estimates.

Comments on the Quality of English Language

Please proofread the text, as there are some sentences that are difficult to follow.

Author Response

Dear reviewer:

We gratefully thank you for your time spend making your constructive remarks and useful suggestions, which has significantly raised the quality of the manuscript and has enables us to improve the manuscript. We have studied your comments carefully and modified the manuscript. Revised portion are highlighted in the paper. The responses to your comments are as following:

  1. Comment: Abstract: A large part of the abstract should be move to introduction. For example, L20-22 is too deep into details about the model, which should not appear in abstract.

Response: Relevant descriptions of model details have been removed from the abstract, and these descriptions of model details have been added to the introduction.

  1. Comment: L17: Is the term "semantic information" appropriately used in this application?

Response: We replace it with "distribution information" in Line 17 so that it can more accurately convey the meaning of the sentence.

  1. Comment: L57-59: It is hard to follow this sentence, revise?

Response: Changes have been made to these two sentences in Line 54-56 so that they may more clearly convey the meaning here.

  1. Comment: L61-76: a lot of proposed methods are mentioned here. However, I am afraid that the readers won't appreciate this section because all these mentions are too brief. For example, what advantages and disadvantages these methods have? Which ones have an overall better performance? These discussions form the foundation of the motivation of this work.

Response: In the Line 59-88 section of the introduction, we add a description of existing research methods and set out their respective strengths. Later we summarize the limitations they share and use this to introduce the methodology of this paper.

  1. Comment: L77: The authors jump directly from ICESat-2 photon denoising into 3D point cloud data denoising, of which the transition is kind of abrupt for the reader. The linkage between the two topics has to be better clarified.

Response: In Line 89-91, some sentences have been added for linking, by the morphological similarity between 3D point cloud data and ICESat-2 photon data, which leads to the existing deep learning based 3D point cloud denoising method.

  1. Comment: L109-110: reference?

Response: Reference has been added here at Line 114.

  1. Comment: L126-128, also Figure 1: Experimental areas A and B seem to be very similar to each other. I suspect that would lead to the overfitting of your model towards this surface type.

Response: This Comment is related to Comment 23, where you might think that this study is using one network model to achieve denoising of data from many different surface types, but in fact this paper trains the model separately for different surface types to achieve a high level of denoising. Thus, in this paper, experimental areas A and B were chosen to test the denoising ability of this paper's method on ICESat-2 data from forested areas.

  1. Comment: L135-137: reference needed.

Response: Relevant references have been added here in Line 142.

  1. Comment: Table 1: table 1 is currently hard to read. I would recommend specifically include dates and times of satellite overpasses in the table.

Response: The acquisition time of each data has been added to Table 1.

  1. Comment: L143-145: DTM and G-LiHT are not defined in the manuscript.

Response: The full names of DTM and G-LiHT have been added in section 2.2.

  1. Comment: Figure 1: you have a figure here showing the location of the 3 experimental areas and the 9 tracks on map. But this figure is not mentioned in the main text.

Response: Section 2.1 of the paper is actually a specific description of Figure 1, and we have included a cue sentence at the beginning of the paragraph to make the object of this description clearer.

  1. Comment: L151: this link is not working.

Response: The link has been corrected, which opens correctly in the browser.

  1. Comment: Figure 2(2): it seems like the "image" data sets were generated by randomly sampling the whole ICESat-2 image. This should be clearly stated in the main text, otherwise it would be difficult to reproduce the results in this paper.

Response: Figure 2(2) represents the photon image transformation step, which is the process of transforming a single photon point and its neighborhood into a 2D image. In this process, all photons of a piece of photon data are transformed into a photon image, where there is no random sampling process. This is mentioned in Line 194. The highlighting of the two photon images in Figure 2(2) is not to indicate that the process is random sampling, but only to show the image conversion process more clearly.

  1. Comment: Figure 2(3): I can understand FC stands for fully-connected layers and what max pooling is. But the authors should not assume that the general audience are familiar with these terms.

Response: Regarding the nomenclature abbreviations in Figure 2(3), the full names and descriptions of these abbreviations have now been added in the last paragraph of Section 3.2 (Line 274-281).

  1. Comment: Figure 2(4): what is "absolute verification"? Do you mean the validation of the elevation estimated by the proposed DL-based method using the independent elevation data sets mentioned in Section 2.2?

Response: You are correct in your understanding that "absolute verification" is the process of validating the elevation of the signal photons extracted by the method in this paper using reference elevation data. We now replace "absolute verification" with "real reference validation" in Fig. 2(4) to be consistent with the later text.

  1. Comment: Figure 3: What is the output of your DL model? Is it a binary output or 2D images? I don't think this schematic diagram is correct.

Response: The DL model in our approach is a binary output model that outputs the class of photon image: signal or noise. Figure 3 does not actually show the output form of the DL model, but rather the specific process of converting from a photon to a photon image. The photon image obtained by this process will be used as an input to the DL model.

  1. Comment: L200-201: The three channels seem to have very different magnitudes. Are any scaling or additional transformation functions applied on these input channels?

Response: Before generating the photon image from the feature matrix, each layer of the feature matrix is subjected to a normalization operation in order to keep the features of each layer in the same order of magnitude. It is now described in Line 235-236.

  1. Comment: L209: references needed here.

Response: Reference has been added here at Line 244.

  1. Comment: L222-231: The authors also need to show a diagram similar to Figs. 3 and 4 for the attention module.

Response: Schematic diagrams of CAM and SAM have been added (Figure 5 and Figure 6) to show specifically the details of the composition of the attention module.

  1. Comment: L242-244: These two sentences should be one sentence.

L246-247: So does it mean that the ground truth is human-labelled? This information should be made clear in the Section 2.

L250-251: hard to follow this sentence

Response: These three comments focus on the process of constructing the simulation data. Now we put this paragraph in "2.3 Training dataset", and the problematic sentences have been modified. Meanwhile, both the training dataset and the validation dataset are indeed manually labeled based on reference data such as topography, as explained in Section 2.3.

  1. Comment: The authors kept using the terms "simulation experiment" and "experimental process", but it turns out to me that they are referring to the training process of the DL model. Please clarify this.

Response: In this paper, "simulation experiment" refers to the process of denoising experiments on simulation data with different denoising methods and comparing the accuracy of the results. In Chapter 4, "experimental process" refers to the execution process and details of all the experiments in this paper.

  1. Comment: L263-265: What does each "piece of data" mean here? Is it referring to the different tracks?

Response: Your understanding is correct, "piece of data" here does refer to data on a track. We have replaced it with "tracks of data" to minimize ambiguity.

  1. Comment: L268-270: If a separate DL model is needed for each surface type, I would say this method is much less useful than a universal DL model that is able to deal with multiple surface types. I also still don't understand why the authors chose to include two areas with almost the same surface features. Is it possible to train the DL model using the combined data set of the three experimental areas?

Response: When we check out the photon data before the experiment began, we noticed that the distribution pattern of photon data in the forested area was different from that in the shallow sea area. The distribution of signal photons in the vegetation canopy in the forest area is very different from that of signal photons at the surface and bottom of the water in the shallow sea area. Meanwhile the distribution pattern of noise photons differs between the two areas, with a more uniform distribution in forested areas and a more skewed distribution toward signal photon aggregation in shallow sea areas. These differences led us to choose to use different models for different surface types. In the end, we also achieved better results for different surface types. We will also actively explore the possibility of training a generalized model in the future.

  1. Comment: Table 2: batch size of 2 is too small and could potentially lead to overfitting.

Response: The maximum batch size here can only be set to 2 due to the limitations of the machine's video memory. If you try to reproduce our experiment, the batch size can be set larger.

  1. Comment: It is surprising to me that it is not mentioned in the manuscript what loss function was used in the training process. If the DL model has binary output format for the classification problem, I'd guess something like cross-entropy was being used here. It is mentioned in L257 that MAE and MRE were used. But what is the exact format of the loss function, as shown by the curves in Figs. 5(b) and 6(b)?

Response: You are correct in your understanding that the loss function we use is the CrossEntropyLoss function, and we have added a description of this in Line 309-310.

  1. Comment: From Figs. 5 and 6, I think overfitting occurred in the training process. It would be great to mention if any efforts are made to avoid overfitting. Also these two figures are displayed but not discussed in the text.

Response: In the process of the experiment, we also found the phenomenon of overfitting, we tried to change the model, expand the training data and other methods, but this problem has not been solved, we speculate that it may be the reason that the batch size is too small. This problem cannot be solved quickly at present. At the same time, we found that the performance of the model obtained from the current training on denoising is currently very good, so we keep the results of the model.

  1. Comment: Figure 7: What is the reasoning behind the design of the size of the window for feature selection?

Response: Since the scale of the photon data is much larger in the along-track distance than in the elevation direction, the window is set to be rectangular in order to allow the window to contain more feature information. The window size is set empirically, and the current window size is 224m*2.24m, which can better preserve the shape features in the photon neighborhood.

  1. Comment: Figure 10(a): Is this validation data set independent from the training process?

Response: This validation dataset is not used in training and is independent of the training process.

  1. Comment: Tables 3, 4 and 5: I think plotting these numbers as curves as functions of SNR would be very helpful to show the robustness of different methods against noise level.

Response: We have plotted these three tables as graphs and placed them in the results presentation (Figure 13, 16, 19 and 22).

  1. Comment: Figures 15 and 16: It would be interesting to see the estimated canopy heights, and (possibly) comparison against some independent estimates.

Response: By now we have not found highly accurate vegetation canopy information within the range of ICESat-2 data we know. Once we have such data, we will continue to do further research.

Round 2

Reviewer 3 Report

Comments and Suggestions for Authors

All my comments have been taken into account, you can publish.

Reviewer 4 Report

Comments and Suggestions for Authors

Thank you for addressing my previous comments.